# Progressive Latent Calibration for Stable Score Distillation

## Abstract

Recent advancements in Score Distillation Sampling (SDS) have significantly accelerated progress in text-to-3D generation by leveraging pre-trained 2D diffusion models to supervise 3D representations. However, SDS often suffers from high variance and produces over-smoothed outputs, limiting the quality of the synthesized 3D assets. While recent methods have introduced DDIM inversion to stabilize optimization, we identify that repeated DDIM inversion introduces discretization errors that accumulate over thousands of iterations, ultimately leading to severe artifacts such as structural distortions and color degradation. To address these limitations, we introduce a novel score distillation framework that eliminates the reliance on DDIM inversion by leveraging multi-step pseudo-ground-truth sampling with progressive latent calibration. Our approach explicitly estimates and reintegrates information loss about the original rendering from a 3D representation during multi-step sampling, thereby preserving semantic fidelity and reducing variance across training iterations. Extensive experiments show that our method consistently outperforms existing inversion-based and standard score distillation approaches in generating high-fidelity 3D assets from text prompts. The anonymous project page is available at https://anonymous-iclr-sd.github.io/.

## 1 Introduction

As 3D computer vision and graphics technologies rapidly advance and gain traction across domains such as gaming, filmmaking, virtual reality (VR), augmented reality (AR), and robotics, the demand for high-quality 3D assets is steadily growing. However, producing a large number of high-quality, hand-crafted 3D assets remains costly and time-consuming, often requiring significant expert effort. Against this backdrop, the field of text-to-3D generation, which enables the creation of desired digital 3D assets from text prompts, is gaining significant attention and growing rapidly. Various branches of approaches (Chen et al., 2023; Hong et al., 2022; Lin et al., 2023; Metzer et al., 2023; Michel et al., 2022; Poole et al., 2023; Wang et al., 2023b; Zhu et al., 2024; Tang et al., 2024; Yi et al., 2024; Jun & Nichol, 2023) have been proposed for text-to-3D generation, but among them, Score Distillation Sampling (SDS) (Poole et al., 2023) has received particular attention as a method that distills the prior knowledge of a pre-trained text-to-image diffusion model to train 3D representations (Mildenhall et al., 2020; Kerbl et al., 2023; Müller et al., 2022) such as Neural Radiance Fields (NeRFs) (Mildenhall et al., 2020) and 3D Gaussian Splatting (3DGS) (Kerbl et al., 2023). This mechanism has provided a breakthrough in overcoming the challenges of the text-to-3D generation task, where constructing large-scale training datasets is often infeasible, and has also started to find applications in a range of downstream tasks such as 3D editing (Hertz et al., 2023; Koo et al., 2024; Nam et al., 2024; Kim et al., 2023), novel view synthesis (Zou et al., 2024; Wang et al., 2024a; Wu et al., 2024), and motion synthesis (Li et al., 2025; Gal et al., 2024).

Despite its popularity and effectiveness, SDS has faced challenges in generating high-fidelity 3D assets due to issues such as over-saturation and over-smoothing, which have limited its practical applicability. These issues in SDS arise from the excessive variance in the update direction at each iteration, caused by applying random noise and performing single-step denoising. As a result, fine-grained features tend to be averaged out during optimization. While Variational Score Distillation (VSD) (Wang et al., 2023b) has addressed these issues and successfully produced high-fidelity, detailed objects, it comes with the drawback of requiring diffusion model fine-tuning at every training iteration, resulting in significantly higher computational costs compared to standard approaches.

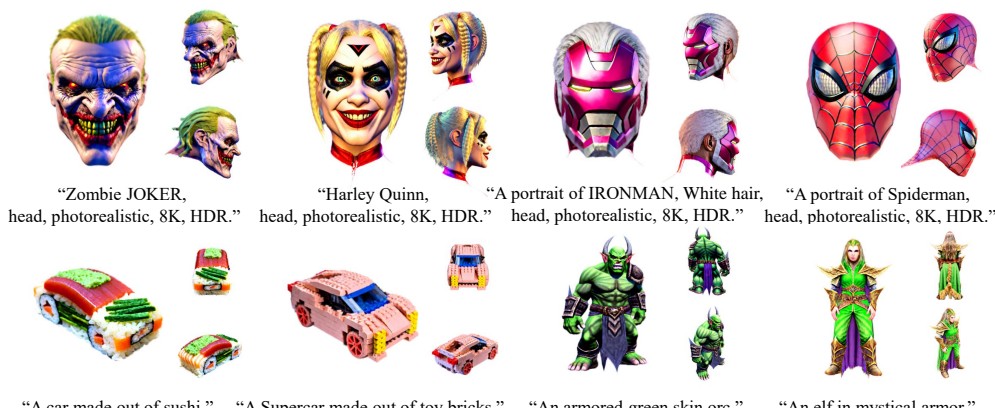

"Zombie JOKER,
head, photorealistic, 8K, HDR."

"Harley Quinn,
head, photorealistic, 8K, HDR."

"A portrait of IRONMAN, White hair,
head, photorealistic, 8K, HDR."

"A portrait of Spiderman,
head, photorealistic, 8K, HDR."

"A car made out of sushi."    "A Supercar made out of toy bricks."    "An armored green skin orc."    "An elf in mystical armor."

Figure 1: Visualization of rendered views from 3D assets generated by the proposed method. Each prompt shown below was used for generation.

To address this issue, recent methods (Liang et al., 2024; Lukoianov et al., 2024; Li et al., 2024c) have explored inversion-based alternatives that leverage DDIM inversion (Song et al., 2021a) to obtain more stable update directions without the need for diffusion model fine-tuning. A notable example, Interval Score Matching (ISM) (Liang et al., 2024) interprets the objective of SDS as matching between the 3D representation's rendering and the diffusion model's predicted sample, which is referred to as the pseudo-ground-truth (pseudo-GT). The authors argue that feature-inconsistent and low-quality pseudo-GTs are a primary cause of the over-smoothing problem in SDS. To address this inconsistency, they introduce DDIM inversion into the score distillation process. Guided Consistency Sampling (GCS) (Li et al., 2024c) and Score Distillation via Inversion (SDI) (Lukoianov et al., 2024) have each contributed to broadening the scope of inversion-based score distillation. Specifically, GCS aims to improve the self-consistency of denoising trajectories within the framework of Probability Flow Ordinary Differential Equations (Song et al., 2021b), while SDI explores the integration of classifier-free guidance (Ho & Salimans, 2022) during DDIM inversion to enhance inversion fidelity.

However, in our analysis, we find that these inversion-based methods suffer from a different set of limitations stemming from the discretization errors (Mokady et al., 2023; Wallace et al., 2023) inherent in the DDIM inversion process. Unlike conventional use cases of DDIM inversion, inversion-based score distillation involves repeatedly generating noisy samples via DDIM inversion and subsequently denoising them to obtain clean samples, spanning from over thousands of iterations. In this setting, even minor deviations at each iteration can accumulate over the distillation process, ultimately leading to artifacts such as structural distortion, color over-saturation, or degradation in the final 3D outputs. Although GCS attempts to mitigate these issues with heuristics such as brightness equalization, this solution is representation-specific and does not generalize well to diverse 3D formats such as NeRF.

To overcome these, we propose a score distillation framework that eliminates the reliance on DDIM inversion while preserving the benefits of consistent and high-quality supervision. By introducing progressive latent calibration into multi-step pseudo-GT sampling, where a corrective noise residual is added at each step of the denoising trajectory, our method generates high-quality and consistent pseudo-GTs. This facilitates the learning of 3D representations with enhanced structural fidelity and color consistency, as shown in Fig. 1. We validate the effectiveness of our method through both quantitative and qualitative comparisons with various state-of-the-art score distillation methods.

## 2 BACKGROUND

### 2.1 DIFFUSION MODELS

Diffusion models (DMs) (Ho et al., 2020; Song et al., 2021b) aim to generate data $x_0$ by progressively denoising a noisy sample $x_t = \alpha_t x_0 + \sigma_t \epsilon$, where $0 < t < T$ and $\alpha_t$, $\sigma_t$ are predefined scheduling functions that satisfy $x_T \approx \epsilon$ and $\alpha_t^2 = 1 - \sigma_t^2$. The forward diffusion process $q(x_t \mid x_{t-1})$ perturbs the data by adding random noise at each timestep. Then, the learned reverse process $p_\phi(x_{t-1} \mid x_t)$ predicts the noise $\epsilon$ from $x_t$ and progressively denoises it to reconstruct the original data $x_0$.

Denoising Diffusion Implicit Models (DDIMs) (Song et al., 2021a) provide an alternative to standard diffusion models by defining a non-Markovian sampling while preserving the marginal distribution of $p(x_t)$. Unlike the original stochastic reverse process in DMs, DDIMs enables a deterministic mapping from $x_T$ to $x_0$. Specifically, starting from a fixed noise $x_T = \epsilon$ and the predicted noise $\epsilon_\phi(x_t, t)$ at each timestep, a deterministic DDIM sampling can be defined as:

$$x_{t-1} = \alpha_{t-1} \left( \frac{x_t - \sigma_t \epsilon_\phi(x_t, t)}{\alpha_t} \right) + \sigma_{t-1} \epsilon_\phi(x_t, t). \tag{1}$$

This deterministic formulation enables the inversion of the sampling process, known as DDIM inversion, where a clean image $x_0$ can be mapped back to its corresponding noise latent $\epsilon_\phi^{\text{inv}}$ by applying the reverse of the DDIM steps. Since this latent noise retains information about $x_0$, DDIM inversion is widely utilized in image editing tasks where preserving the coarse structure of the input is crucial (Su et al., 2022; Mokady et al., 2023; Wallace et al., 2023).

## 2.2 SCORE DISTILLATION

With the advancement of diffusion models, various methods (Poole et al., 2023; Wang et al., 2023b; Liang et al., 2024; Lukoianov et al., 2024; Li et al., 2024c; Zhuo et al., 2024) have been proposed to leverage pre-trained 2D text-to-image diffusion models for distilling into parameterized 3D representations, such as NeRF (Mildenhall et al., 2020; Müller et al., 2022) and 3DGS (Kerbl et al., 2023). These approaches follow the initial formulation introduced by Score Distillation Sampling (SDS) (Poole et al., 2023). SDS renders a 3D representation $g_\theta$ at each iteration using randomly sampled conditions $c$, including camera parameters, background, and lighting conditions. The rendered sample $x_0 = g_\theta(c)$ is then perturbed with random noise $\epsilon$, and a pre-trained diffusion model $\phi$ is used to predict the $\epsilon$ from the noisy sample $x_t$. The 3D representation is optimized as below using the objective function $\mathcal{L}_{\text{SDS}}$ based on the noise residual

$$\min_\theta \mathcal{L}_{\text{SDS}}(\theta) = \mathbb{E}_{t,c} \left[ \omega(t) \left\| \hat{\epsilon}_\phi(x_t; t, y) - \epsilon \right\|_2^2 \right], \tag{2}$$

where $y$ denotes a text prompt, $\omega(t)$ is a weighting function, and $\hat{\epsilon}_\phi$ denotes the noise prediction guided by classifier-free guidance (CFG) (Ho & Salimans, 2022) with prompt $y$. As demonstrated in prior works (Zhu et al., 2024; Liang et al., 2024), the objective function can be equivalently expressed in terms of the sample residual

$$\min_\theta \mathcal{L}_{\text{SDS}}(\theta) = \mathbb{E}_{t,c} \left[ \lambda(t) \left\| x_0 - \hat{x}_0^{\phi,y} \right\|_2^2 \right], \tag{3}$$

where $\hat{x}_0^{\phi,y} = (x_t - \sigma_t \hat{\epsilon}_\phi(x_t; t, y))/\alpha_t$ denotes a sample predicted by DMs, and $\lambda(t) = \omega(t) \cdot \alpha_t/\sigma_t$. Therefore, SDS can be interpreted as optimizing $g_\theta$ such that the rendered view $x_0$ matches the DM's prediction $\hat{x}_0^{\phi,y}$; as a result, $\hat{x}_0^{\phi,y}$ is often referred to as a pseudo-ground-truth (pseudo-GT) (Liang et al., 2024). SDS has established the basis for text-to-3D generation using pre-trained 2D DMs and has shown strong potential for practical use. However, SDS produces highly variant pseudo-GTs at each training iteration, as $\hat{x}_0^{\phi,y}$ varies according to added random noise, leading to the over-smoothing problem (Liang et al., 2024). To mitigate this, a large CFG weight (e.g., around 100) is commonly used, at the cost of introducing over-saturation and low diversity.

A notable approach improving SDS is Variational Score Distillation (VSD) (Wang et al., 2023b), which successfully generates high-detail 3D objects under standard CFG scales. VSD introduces another diffusion model $\psi$, which is initialized with $\phi$ and fine-tunes it via LoRA (Hu et al., 2022) to approximate the current state $\hat{x}_0^\psi$ from $x_t$ during score distillation. Instead of directly optimizing $g_\theta$ with the residual of the clean sample $x_0$ and pseudo-GT $\hat{x}_0^\phi$, VSD utilizes the residual between $\hat{x}_0^\psi$ and $\hat{x}_0^\phi$. As $\hat{x}_0^\psi$ estimates the score of the noisy $x_t$, VSD effectively updates 3D objects with the random noise. However, the additional LoRA fine-tuning introduces severe computational overhead.

Recently, several methods have been proposed to mitigate the computational burden and address the high variance issue in score distillation by leveraging DDIM inversion (Song et al., 2021a). Interval Score Matching (ISM) (Liang et al., 2024) highlights that single-step denoising often produces blurry outputs and thus proposes a new objective that combines a multi-step prediction with DDIM inversion using a null text prompt. Score Distillation via Inversion (SDI) (Lukoianov et al., 2024)

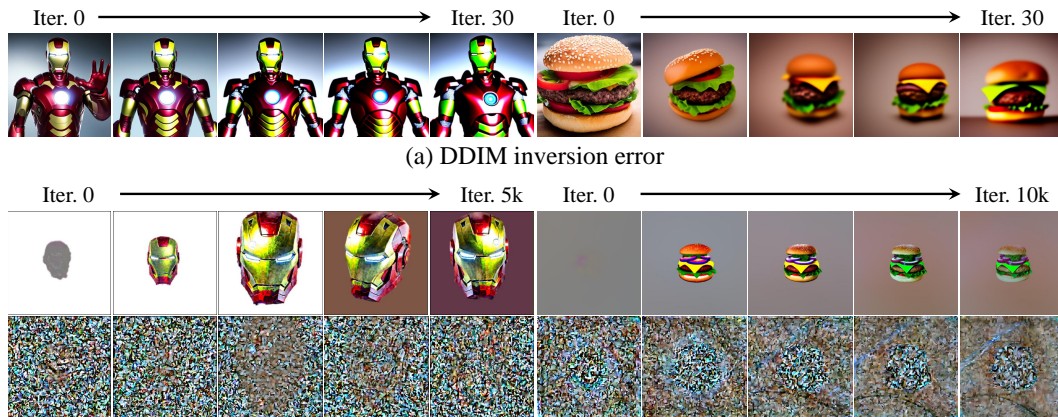

Iter. 0 ⟶ Iter. 30    Iter. 0 ⟶ Iter. 30

(a) DDIM inversion error

Iter. 0 ⟶ Iter. 5k    Iter. 0 ⟶ Iter. 10k

(b) Artifacts of inversion-based score distillation

Figure 2: (a) We recursively apply 50-step (left) and 10-step (right) DDIM inversion to the input image, followed by 50-step DDIM sampling for reconstruction using StableDiffusion 2.1 (Rombach et al., 2022). The accumulation of inversion errors can lead to structural distortions or blurring, as well as color oversaturation or degradation. (b) Inversion errors in pseudo-GTs accumulate during iterative score distillation, resulting in oversaturation (left, ISM (Liang et al., 2024)) or color degradation and structural distortions (right, SDI (Lukoianov et al., 2024)). The predicted noise (bottom) also deviates from a normal distribution, providing further evidence of error accumulation in DDIM inversion.

replaces the null prompt used in DDIM inversion with a text prompt combined with a negative CFG scale. Guided Consistency Sampling (GCS) (Li et al., 2024c) modifies the ISM objective by using a midpoint timestep, addressing the increased error from single-step denoising at large $t$. Additionally, it proposes the compact consistency loss to improve the self-consistency of the denoising trajectory defined by the probability flow ordinary differential equations (PF-ODEs).

Although inversion-based methods appear to successfully generate high-fidelity 3D assets within a reasonable computational budget, they often struggle to produce accurate colorization due to the accumulation of discretization errors inherent in the DDIM inversion process. In particular, ISM and SDI suffer from color oversaturation and color degradation, respectively. To address the oversaturation issue, GCS introduces a brightness-equalized generation (BEG) strategy that resets the brightness of Gaussian splatting. However, this approach is limited in its applicability to other 3D representations such as NeRF. We further discuss this issue in detail in the following section.

## 3 METHOD

In this section, we analyze how DDIM inversion, as employed by various recent score distillation methods (Liang et al., 2024; Lukoianov et al., 2024; Li et al., 2024c), negatively impacts the generation of 3D assets, leading to issues such as color degradation, over-saturation, and structural distortions. Then, we propose a novel score distillation method that eliminates the need for DDIM inversion by progressively calibrating the diffusion sampling trajectory to align with the latent space of the original rendering, addressing the high variance issue of SDS while avoiding the artifacts from inversion.

### 3.1 ANALYSIS OF DDIM INVERSION IN SCORE DISTILLATION

DDIM inversion (Song et al., 2021a;b) reconstructs the latent noise $\epsilon_\phi^{\text{inv}}$ by recursively adding predicted noise to the rendered image. Ideally, $\epsilon_\phi^{\text{inv}}$ should follow a Gaussian distribution. However, due to discretization errors inherent in DDIM sampling, the inversion trajectory gradually deviates from the forward diffusion trajectory, leading to a mismatch between $\epsilon_\phi^{\text{inv}}$ and the ideal Gaussian noise. This distributional discrepancy can hinder the accurate reconstruction of the original image (Mokady et al., 2023; Wallace et al., 2023; Ju et al., 2023). While such inversion errors are often negligible—typically observed as minor color shifts or local structural artifacts—they become significantly amplified in the context of score distillation, where the inversion process is applied thousands of times throughout

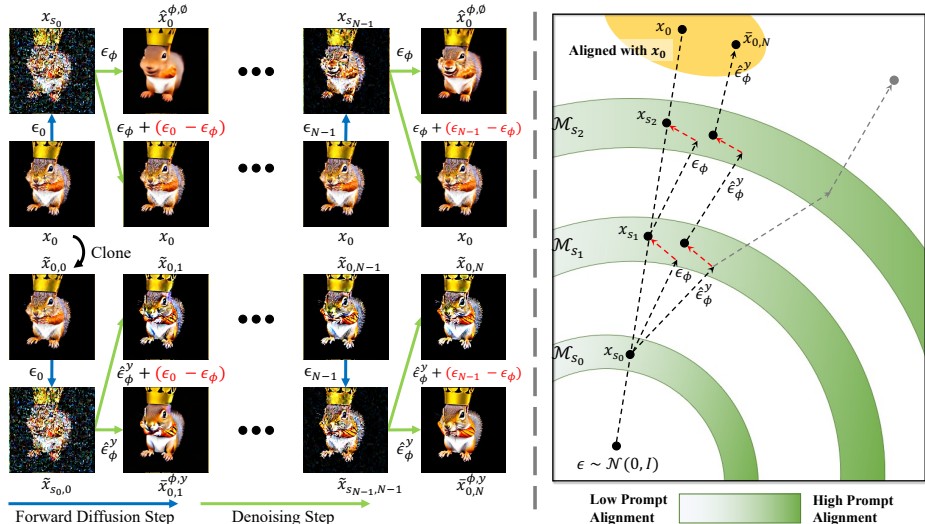

Figure 3: **Overview of the proposed method.** (Left) Visualization of the proposed multi-step sampling strategy. (Right) The conceptual illustration of the progressive latent calibration with diffusion latent manifold ($\mathcal{M}$). At each sampling step, we explicitly preserve the information of $x_0$ by adding the residual between random noise $\epsilon$ and unconditional prediction $\epsilon_\phi$ to CFG prediction $\hat{\epsilon}_\phi^y$, which is denoted as a red dotted arrow. Without this, the sampling trajectory often deviates from $x_0$, generating unaligned pseudo-GTs and causing artifacts (gray dotted arrow). For simplicity, the conceptual illustration depicts the noise for each iteration $\epsilon_n$ as fixed. Best viewed in color.

iterative optimization, leading to substantial degradation in the resulting 3D assets. Moreover, due to the computational constraints, inversion-based score distillation methods often adopt a few-step inversion (generally less than 10 steps), which further amplifies inversion errors.

To investigate the effect of the accumulated DDIM inversion error, we conduct a 2D toy experiment in Fig. 2 (a). Given an input image, we iteratively apply unconditional DDIM inversion (Mokady et al., 2023) and reconstruct images with a CFG scale of 7.5. The results demonstrate that even minor inversion errors accumulate over successive inversion and generation processes, leading to noticeable artifacts. Accordingly, pseudo-GTs generated from latent noise $\epsilon^\phi$ inevitably include DDIM inversion errors, which propagate and accumulate in the resulting 3D assets through $\mathcal{L}_{SDS}$ during optimization as shown in Fig. 2 (b). A potential remedy is to use improved inversion techniques that reduce discretization errors (Elarabawy et al., 2022; Wallace et al., 2023). However, the limited number of inversion steps remains a fundamental constraint, making it difficult to fully eliminate inversion artifacts. To this end, we design an inversion-free score distillation method that produces consistent pseudo-GTs without requiring inversion.

## 3.2 PROGRESSIVE LATENT CALIBRATION FOR CONSISTENT PSEUDO-GTS

Our approach aims to sample high-quality pseudo-GTs from $x_t$ that are aligned with $x_0$ to provide effective supervision for training the 3D representation. In particular, we utilize $x_t$ sampled with random noise $\epsilon$ to avoid error accumulation introduced by DDIM inversion. As mentioned in ISM (Liang et al., 2024), a single denoising step in SDS tends to produce low-quality pseudo-GTs. To alleviate this, we consider a multi-step sampling strategy that performs a total of $N$ iterative updates for generating pseudo-GTs. Let $n \in [0, N-1]$ be the sampling step index, then the sample obtained through naïve multi-step CFG sampling $\acute{x}_{0,n+1}$ can be calculated as:

$$\acute{x}_{0,n+1} = \frac{\acute{x}_{s_n,n} - \sigma_{s_n}\left(\hat{\epsilon}_\phi(\acute{x}_{s_n,n}; s_n, y)\right)}{\alpha_{s_n}},  \tag{4}$$

where the diffusion sampling timestep of $n$-th update step is $s_n$, and $\acute{x}_{i,j}$ denotes the noisy sample at timestep $i$ and sampling step $j$. While multi-step CFG sampling can produce high-quality pseudo-GTs that align well with the input prompt, it still suffers from unstable pseudo-GTs due to the high

variance of random noise. As a result, despite its visual superiority, it still inherits the problem of SDS. Therefore, we propose a modified multi-step sampling strategy with CFG that maintains high quality and prompt alignment, while encouraging the $n$-th step sample $\tilde{x}_{0,n}$ to stay aligned with $x_0$.

During the CFG sampling step from $\tilde{x}_{0,n}$, adding noise and subsequently reconstructing the sample through the DM's prior inevitably leads to partial loss of information related to the original rendering $x_0$ from $\tilde{x}_{0,n}$. Since $\tilde{x}_{0,n}$ has already undergone multiple update steps from $x_0$, directly recovering this information loss with respect to $x_0$ is highly challenging. We design a method that explicitly estimates the lost information by computing the difference between the actual noise and the unconditionally predicted noise inferred from the DM's prior knowledge. More specifically, at the $n$-th step, we first generate a noisy sample $x_{s_n} = \alpha_{s_n} x_0 + \sigma_{s_n} \epsilon_n$ by adding noise $\epsilon_n \sim \mathcal{N}(0, I)$ directly to $x_0$. We then apply an unconditional noise prediction step to obtain the noise $\epsilon_\phi(x_{s_n}; s_n, \emptyset)$ from $x_{s_n}$. Each addition of random noise $\epsilon_n$ during the multi-step diffusion sampling increasingly removes specific details of the original sample $x_0$. Accordingly, the unconditional noise prediction $\epsilon_\phi(x_{s_n}; s_n, \emptyset)$, guided solely by the DM's learned prior, provides a baseline estimate of missing information based only on general knowledge of the data distribution. Therefore, the residual between the actual noise $\epsilon_n$ and the unconditional prediction serves as an estimate of which specific aspects of $x_0$ are no longer preserved in $\tilde{x}_{0,n}$ due to the addition of $\epsilon_n$. By incorporating this estimated residual information back into the conventional CFG sampling step, we explicitly reintroduce an approximation of the lost original details, thus potentially improving the preservation of $x_0$-specific semantic and visual features within $\tilde{x}_{0,n+1}$. We refer to this process, in which the lost information is recovered to calibrate the latent towards $x_0$, as progressive latent calibration. The sample $\tilde{x}_{0,n+1}$ obtained by the modified sampling at $n$-th step can be formulated as:

$$\tilde{x}_{0,n+1} = \frac{\tilde{x}_{s_n,n} - \sigma_{s_n}\left(\hat{\epsilon}_\phi(\tilde{x}_{s_n,n}; s_n, y) + \epsilon_n - \epsilon_\phi(x_{s_n}; s_n, \emptyset)\right)}{\alpha_{s_n}}, \tag{5}$$

where $\tilde{x}_{i,j} = \alpha_i \tilde{x}_{0,j} + \sigma_i \epsilon_j$. We visualize the multi-step sampling strategy with the proposed progressive latent calibration (PLC) in Fig. 3, and we further provide the intuition and theoretical analysis for the residual correction term from the guidance perspective in Appendix I.

We generate the pseudo-GT $\bar{x}_{0,N}^{\phi,y}$ by applying the proposed progressive latent calibration to multi-step CFG sampling, which serves as the supervision target for the matching loss with $x_0$. For the initial sampling step, $\tilde{x}_{0,0}$ is initialized with $x_0$. After $(N-1)$ steps of sampling with latent calibration, we can obtain a consistent and high-quality predicted sample $\tilde{x}_{0,N-1}$. Then, pseudo-GT $\bar{x}_{0,N}^{\phi,y}$ is derived by applying one final CFG sampling step, where $\bar{x}_{0,n}^{\phi,y} = (\tilde{x}_{s_{n-1},n-1} - \sigma_{s_{n-1}}\hat{\epsilon}_\phi(\tilde{x}_{s_{n-1},n-1}; s_{n-1}, y))/\alpha_{s_{n-1}}$.[1] The objective from a matching loss term $\mathcal{L}_m$ between the generated pseudo-GT and the original rendering $x_0$ can be formulated as:

$$\min_\theta \mathcal{L}_m(\theta) = \mathbb{E}_{t,c}\left[\lambda(t)\left\|x_0 - \bar{x}_{0,N}^{\phi,y}\right\|_2^2\right], \tag{6}$$

where $t$ is equal to $s_0$, the timestep of the initial sampling step. By leveraging this, we provide consistent and high-quality supervision to the 3D representation, which in turn stabilizes the score distillation process.

In addition to the $\mathcal{L}_m$ term that aims to match the original rending with the consistent and high-quality pseudo-GT, we introduce an auxiliary loss $\mathcal{L}_{aux}$ that incurs no additional computational overhead. This auxiliary loss encourages the unconditional prediction $\bar{x}_{0,2}^{\phi,\emptyset}$ after the first update step of each iteration, $\tilde{x}_{s_1,1}$, to match the CFG prediction derived from the original rendering $\hat{x}_0^\phi$. By doing so, it guides the update direction of 3D representations to remain aligned with the prompt-aligned prediction, thereby improving text alignment during the learning of the 3D representation. The auxiliary loss objective can be expressed as follows:

$$\min_\theta \mathcal{L}_{aux}(\theta) = \mathbb{E}_{t,c}\left[\lambda(t)\left\|\hat{x}_0^{\phi,y} - \bar{x}_{0,2}^{\phi,\emptyset}\right\|_2^2\right]. \tag{7}$$

Accordingly, the overall training objective of our method $\mathcal{L}_{total}$ combines both the pseudo-GT matching loss and the auxiliary loss, formulated as their weighted sum as follows: $\mathcal{L}_{total} = \mathcal{L}_m + w * \mathcal{L}_{aux}$, where $w$ is a weight hyperparameter for the auxiliary loss.

---

[1]Note that the final sampling step does not incorporate latent calibration as directly calibrating in the clean image space may lead to artifacts; therefore, the proposed method is dubbed as *latent* calibration.

Table 1: **Quantitative results.** We evaluate state-of-the-art score distillation methods in terms of CLIP score, FID and runtime.

|  | SDS | VSD[†] | SDI[†] | ISM | GCS | Ours |
|---|---|---|---|---|---|---|
| CLIP score ↑ | 30.68 | **33.17** | 27.15 | 31.30 | 32.37 | 32.68 |
| FID ↓ | 137.22 | 119.53 | 231.24 | 109.57 | 103.40 | **100.41** |
| Runtime (min) | 32 | 254 | 162 | 128 | 168 | 133 |

[†] As VSD and SDI exhibit high VRAM consumption beyond the capacity of an RTX 4090, their runtime was measured on an A6000.

Table 2: User preference test results.

|  | ISM | GCS | Ours |
|---|---|---|---|
| Q1 | 12.7% | 23.3% | 64.0% |
| Q2 | 14.7% | 27.3% | 58.0% |
| Q3 | 16.7% | 24.0% | 59.3% |

## 4 EXPERIMENTS

### 4.1 IMPLEMENTATION DETAILS

Our method is implemented on top of the ISM (Liang et al., 2024) codebase, and we follow its settings for hyperparameters such as camera parameters and learning rate. The 3D Gaussian initialization is conducted using Point-e (Nichol et al., 2022). The CFG scale is fixed to 7.5 and we use Stable Diffusion 2.1 Base (Rombach et al., 2022) as a base diffusion model across all experiments. Each asset is trained for 5,000 iterations. The auxiliary loss weight $w$ is set to 0.8. We sample the timestep $t$ uniformly between a minimum and a maximum timestep, where the minimum timestep decreases from 500 to 151 and the maximum timestep from 980 to 500 over a warm-up period of 1,500 iterations. After the warm-up period, $t$ was uniformly sampled from the interval $[151, 500]$. We set the number of the pseudo-GT sampling steps $N$ to 4. Specifically, we define $s_0 = t$, $s_1 = t - 50$, and sample $s_{N-1}$ uniformly from the interval $[20, 100]$. The remaining intermediate timesteps are linearly interpolated between $s_1$ and $s_{N-1}$. See Appendix A for further details.

### 4.2 EXPERIMENTAL RESULTS

**Quantitative results.** To evaluate the proposed method in terms of prompt alignment and overall quality, we measure the CLIP score (Radford et al., 2021) and FID (Heusel et al., 2017), and compare the results against state-of-the-art score distillation variants, including SDS (Poole et al., 2023), VSD (Wang et al., 2023b), and various inversion-based methods (Lukoianov et al., 2024; Liang et al., 2024; Li et al., 2024c). We adopt prompts from the Dreamfusion gallery (Poole et al., 2022) following the recent state-of-the-art method (Li et al., 2024c). The CLIP score is computed using the CLIP ViT-B/16 model, and we follow the evaluation protocol of previous work (Poole et al., 2023). To measure FID, we generate 50,000 images from the 30 evaluation prompts using Stable Diffusion 2.1 Base (Rombach et al., 2022). These images are then used to compute FID against the renderings from our 3D assets, where each of the 30 assets is rendered from 120 different views. This setup allows us to assess how well our method distills the generative capability of the base diffusion model. We report the average runtime of the methods using an RTX4090 to measure the efficiency.

As shown in the Table 1, the quantitative results demonstrate that the proposed method achieves prompt alignment comparable to that of VSD, which yields the highest CLIP score, while significantly reducing computational cost and improving generation quality. Moreover, the best FID score indicates that our method most effectively distills the generative capability of the base diffusion model, and that the resulting 3D assets exhibit high-fidelity geometry and appearance. Although our method adopts a multi-step sampling strategy to preserve fine details in the generated 3D assets, it does not require the diffusion inference steps for DDIM inversion. As a result, the average runtime shows that our method achieves comparable or even lower computational cost than most inversion-based methods.

**Qualitative results.** As quantitative metrics cannot fully capture the semantic coherence perceived by humans, we further provide qualitative results of the generated 3D assets. The first (IRONMAN) and third (hamburger) row of Fig. 4 illustrates that DDIM inversion-based methods exhibit pronounced color distortions in the generated 3D assets, a phenomenon that stems from the error accumulation problem identified in Section 3.1. Furthermore, SDI consistently exhibits color degradation across all samples. In contrast, our method eliminates DDIM inversion during the distillation process, thereby avoiding artifacts caused by accumulated errors and maintaining natural color without relying on any additional RGB loss. In addition, our method produces highly detailed photo-realistic results with well-preserved textures across a wide range of prompts, as it provides supervision from high-quality and consistent pseudo-GTs during the distillation of the 3D representation.

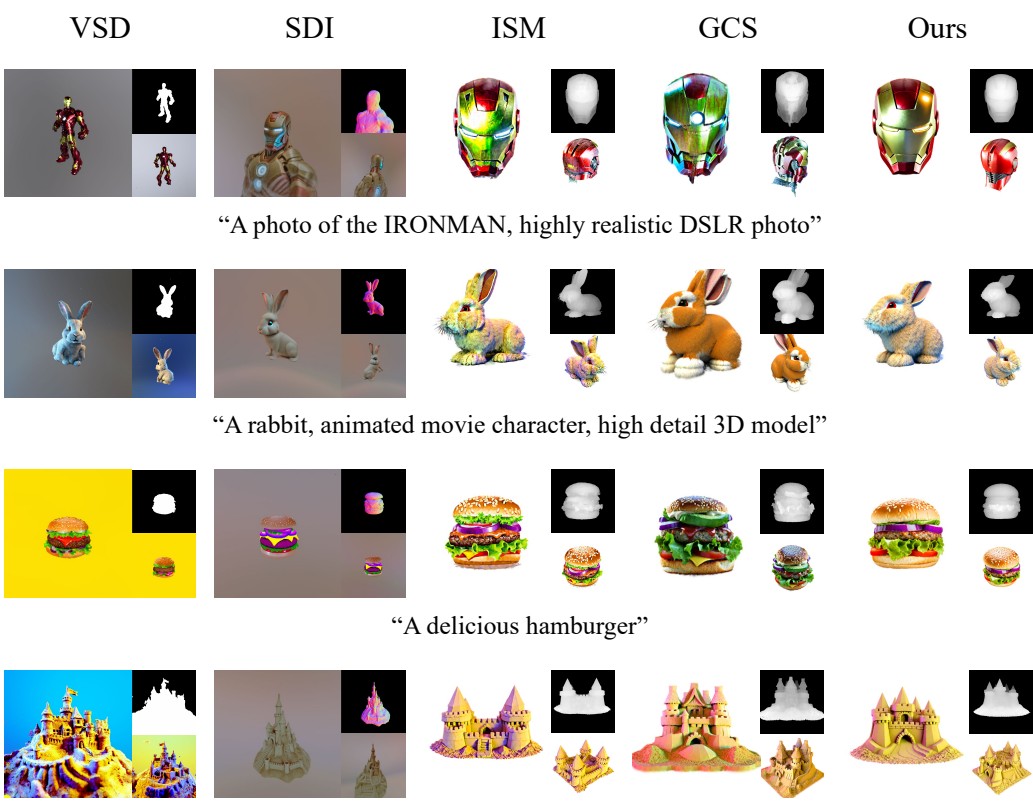

Figure 4: **Qualitative results.** We visualize the 3D assets generated from four prompts using state-of-the-art score distillation methods as well as our proposed method. For each prompt and method, we provide the main view, the corresponding opacity / normal / depth map of the main view, and an additional sub-view. Best viewed in close-up; see Appendix O for details.

### 4.3 USER PREFERENCE TEST

To evaluate user preference between the proposed method and 3D Gaussian Splatting-based score distillation methods, ISM and GCS, we conducted a user study. Following the user preference test setup from previous work (Li et al., 2024c), we used the same 30 3D assets employed to measure the quantitative results and surveyed 30 participants. For each participant, $360°$ rendered videos of objects created using 3 different methods for 5 randomly selected prompts were presented. The participants were then asked to choose the preferred result among the methods for following questions:

- Q1: *Which of the methods produces 3D assets with the highest overall quality (e.g., realism, detail, visual appeal)?*
- Q2: *Which of the following methods aligns most with the text prompt?*
- Q3: *Which of the methods produces 3D assets with the most natural and realistic color appearance?*

In Table 2, our method achieved the highest user preference scores for all questions, receiving 64.0% for Q1, 58.0% for Q2, and 59.3% for Q3. These results indicate that our approach is most favored in terms of overall quality, prompt alignment, and color naturalness among the compared methods.

### 4.4 ABLATION STUDY

**Number of sampling steps.** We conducted an ablation study to investigate the effect of the number of sampling steps $N$, a hyperparameter in our method. As shown in Fig. 5, our method consistently generates high-quality 3D assets across all tested values $N = 3, 4, 5$, indicating that the overall generation quality is insensitive to the choice of $N$. Nonetheless, closer inspection reveals that when

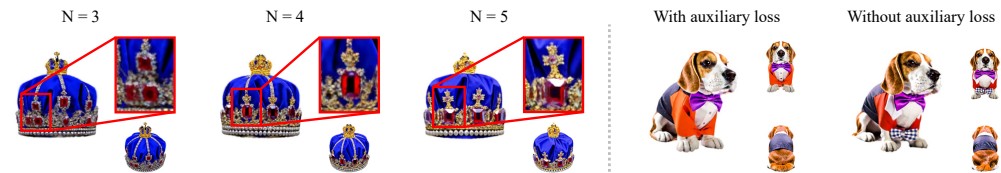

Figure 5: **Ablation study on the number of sampling steps and auxiliary loss.** The samples visualize the effects of varying the number of sampling steps and the presence of the auxiliary loss. The prompts *"a DSLR photo of the Imperial State Crown of England"* and *"a beagle in a detective outfit"* were used for the respective ablation studies.

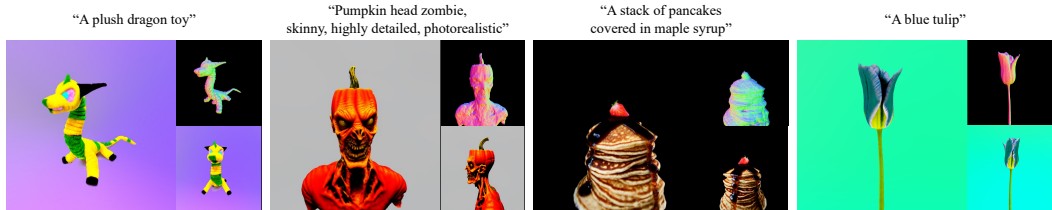

Figure 6: **Ablation study on the 3D representations.** We replaced the original 3D representation, 3D Gaussian Splatting, with Instant-NGP and visualized the generation results for four prompts.

$N = 3$, certain fine-grained details are less faithfully generated compared to higher values. When $N = 4$ or 5, fine details are well preserved with minimal differences across samples. However, increasing $N$ also leads to a rise in computational cost, with total runtimes of 104, 133, and 150 minutes for $N = 3, 4, 5$, respectively. We report more detailed ablation results in Appendix L.

**Loss fuction.** We conducted an ablation study to evaluate the impact of the proposed auxiliary loss term $\mathcal{L}_{aux}$. As shown in Fig. 5, even without the auxiliary loss, the pseudo-GT matching loss can still guide the generation of detailed 3D assets by aligning high-quality and consistent pseudo-ground truths with the renderings. However, we observe that class-discriminative features appear less distinct when the auxiliary loss is omitted. This indicates that the auxiliary loss aids effectively in improving prompt alignment, particularly by enhancing the consistency of class-specific details with the prompt.

**3D representations.** To demonstrate that our proposed method is broadly applicable beyond 3DGS, we conducted an ablation study by replacing 3DGS with InstantNGP (Müller et al., 2022), a NeRF-based variant. The implementation was based on the open-source project Threestudio (Guo et al., 2023), and we followed the configuration of SDI (Lukoianov et al., 2024). As shown in Fig. 6, our method successfully generates high-quality 3D assets with fine details even when using InstantNGP.

## 4.5 COLOR FIDELITY AND OVER-SATURATION ANALYSIS

Standard quantitative metrics such as FID and CLIP Score are often insufficient to effectively capture and quantify persistent color artifacts (e.g., over-saturation and color distortion) in 3D assets generated by score distillation. To objectively prove the improvement in color fidelity, which is a key benefit of our framework, we adopt a dedicated evaluation metric. To directly assess the color improvement, we measure the saturation score following Sadat et al. (2025).

Specifically, it first converts each image from RGB to HSV color space and computes the mean of the saturation channel. Then the RMS contrast is calculated as the standard deviation of pixel values after converting the image to grayscale. The final metrics are derived by averaging the saturation and RMS contrast values across all images. We utilize the same images utilized for FID and CLIP score measuring in the main experiment, in addition to the 10k images from COCO validation set Lin et al. (2014). We exclude the background of 3D assets for calculating metrics.

Table 3 clearly shows that compared to GCS and ISM, our method shows closer saturation and contrast with real images and SD-generated images. While SDI shows a closer saturation score to

Table 3: **Quantitative analysis of color fidelity and contrast metrics.** For each metric, we additionally report its difference from real images.

| Method | Saturation | Contrast |
|---|---|---|
| COCO-val (Real Images) | 0.319 | 0.232 |
| Generated Samples w/ SD 2.1 | 0.336 (+0.017) | 0.234 (+0.002) |
| GCS (Li et al., 2024c) | 0.505 (+0.186) | 0.245 (+0.013) |
| ISM (Liang et al., 2024) | 0.522 (+0.203) | 0.248 (+0.016) |
| SDI Lukoianov et al. (2024) | 0.362 (+0.043) | 0.140 (-0.092) |
| Ours | 0.439 (+0.120) | 0.221 (-0.011) |

real images, it shows a significantly lower contrast score compared to real images. This is due to severe color degradation as demonstrated in Fig. 4. Overall, these quantitative metrics confirm that PLC successfully achieves the best balance of color fidelity and contrast among all tested methods.

## 5 CONCLUSION

In this work, we examine how repeated DDIM inversion in inversion-based score distillation methods leads to artifacts in the resulting 3D assets. Furthermore, we propose a multi-step pseudo-GT sampling strategy using progressive latent calibration that mitigates the high variance issue in score distillation sampling without relying on DDIM inversion. Building on this, we define a new score distillation objective that overcomes the limitations of existing methods. Through extensive experiments and analyses, we demonstrate that our method can effectively generate high-fidelity 3D assets from text prompts.

**Limitations.** While PLC substantially alleviates the limitations commonly observed in score distillation methods, it is not free from multi-view inconsistencies such as the Janus problem (Armandpour et al., 2023), and occasionally fails to fully capture complex compositional prompts. See Appendix E for more comprehensive analysis with examples.

## ETHICS STATEMENT

The generation of realistic 3D assets from text prompts, as investigated in our work, presents important ethical considerations. Like other generative models, it could be potentially misused by unauthorized users. Potential risks include the creation of deceptive content and the generation of misleading media, which can impact individual privacy and public trust. Our method's reliance on 2D diffusion models means it inherits their capabilities and limitations, including potential biases and harmful contents present in the training data. To mitigate misuse, we have based our work on public, well-established 2D generative models that incorporate safety filters. We also acknowledge that biases from the underlying datasets may be reflected in the generated 3D assets.

This work is intended solely for academic research and constructive applications. We strictly prohibit malicious or unethical uses of the proposed method and strongly emphasize the need for clear ethical guidelines to ensure the responsible advancement of the 3D generative research community.

## REPRODUCIBILITY STATEMENT

We have made extensive efforts to ensure the reproducibility of our work. Details of the proposed PLC framework and the new score distillation objective are fully described in the main manuscript. Comprehensive training and evaluation settings are provided in section 4.1 and further elaborated in Appendix A. All model weights for 2D diffusion and the prompts used for evaluation are clearly documented and publicly available. In addition, we provide further ablation studies and qualitative results in the supplementary materials, along with an anonymous website that comprehensively visualizes the quality of the generated 3D assets.

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

APPENDIX

## A  FURTHER DETAILS

For certain prompts, we employed Perp-Neg (Armandpour et al., 2023) to improve multi-view consistency. Additionally, we incorporated the negative prompts proposed in NFSD (Katzir et al., 2024) within the diffusion sampling procedure, leveraging their demonstrated ability to improve sample quality by suppressing unwanted features. We did not apply the brightness reset technique suggested by GCS (Li et al., 2024c), as our method inherently avoids inversion-related artifacts that brightness reset aims to mitigate. For camera parameters, we strictly followed the conventions established by ISM (Liang et al., 2024), ensuring alignment with their setup and facilitating fair comparison. Following GCS, we also incorporate gradient clipping (Pan et al., 2023) for our PLC framework. While gradient clipping helps mitigate the over-saturation problem, it does not fully address the problem, as discussed in appendix J.

## B  ALGORITHM

---

**Algorithm 1:** Multi-Step Sampling With Progressive Latent Calibration

---

**Input:** Initial latent $\mathbf{x}$, timestep set $\mathcal{T}$, weighting functions $\alpha_t(\cdot)$, $\sigma_t(\cdot)$, CFG scale $s$, noise ratio $\eta$, text prompt $y$, number of steps $N$, diffusion model weight $\phi$, auxiliary loss weight $\lambda$

**Output:** $\bar{\mathbf{x}}$, $\bar{\mathbf{x}}_{\text{aux}}$, $\hat{\mathbf{x}}$

1   Sample terminal noise $\boldsymbol{\epsilon}_T \sim \mathcal{N}(0, \mathbf{I})$ ;

2   Initialize: $\tilde{\mathbf{x}} \leftarrow \mathbf{x}$ ;

3   **for** $i = 0$ **to** $N - 1$ **do**

4      $t_i \leftarrow \mathcal{T}[i]$ ;

5      $\alpha \leftarrow \alpha_t(t_i), \quad \sigma \leftarrow \sigma_t(t_i)$ ;

6      Sample $\boldsymbol{\eta} \sim \mathcal{N}(0, \mathbf{I})$;

     // Generate noisy latents

7      $\mathbf{x}_t \leftarrow \alpha \cdot \mathbf{x} + \sqrt{(1 - \eta) \cdot \sigma^2} \cdot \boldsymbol{\epsilon}_T + \sqrt{\eta \cdot \sigma^2} \cdot \boldsymbol{\eta}$;

8      $\tilde{\mathbf{x}}_t \leftarrow \alpha \cdot \tilde{\mathbf{x}} + \sqrt{(1 - \eta) \cdot \sigma^2} \cdot \boldsymbol{\epsilon}_T + \sqrt{\eta \cdot \sigma^2} \cdot \boldsymbol{\eta}$;

     // Predict noise for noisy latents

9      $\boldsymbol{\epsilon} \leftarrow \frac{\mathbf{x}_t - \alpha \cdot \mathbf{x}}{\sigma}$;

10      $(\boldsymbol{\epsilon}_{\text{uncond}}, \tilde{\boldsymbol{\epsilon}}_{\text{cfg}}, \tilde{\boldsymbol{\epsilon}}_{\text{uncond}}) \leftarrow \texttt{PredictNoise}(\phi, \mathbf{x}_t, \tilde{\mathbf{x}}_t, t_i, y, s)$;

11      $\boldsymbol{\epsilon}_T \leftarrow \tilde{\boldsymbol{\epsilon}}_{\text{cfg}} + \boldsymbol{\epsilon} - \boldsymbol{\epsilon}_{\text{uncond}}$;

     // Latent update

12      $\tilde{\mathbf{x}} \leftarrow \frac{\tilde{\mathbf{x}}_t - \sigma \cdot \boldsymbol{\epsilon}_T}{\alpha}$;

     // Store $\hat{\mathbf{x}}$, $\bar{\mathbf{x}}_{\text{aux}}$ for auxiliary loss

13      **if** $i = 0$ **then**

14         $\hat{\mathbf{x}} \leftarrow \frac{\tilde{\mathbf{x}}_t - \sigma \cdot \tilde{\boldsymbol{\epsilon}}_{\text{cfg}}}{\alpha}$;

15      **if** $i = 1$ **then**

16         $\bar{\mathbf{x}}_{\text{aux}} \leftarrow \frac{\tilde{\mathbf{x}}_t - \sigma \cdot \tilde{\boldsymbol{\epsilon}}_{\text{uncond}}}{\alpha}$;

17   $\bar{\mathbf{x}} \leftarrow \frac{\tilde{\mathbf{x}}_t - \sigma \cdot \tilde{\boldsymbol{\epsilon}}_{\text{cfg}}}{\alpha}$;

18   **return** $\bar{\mathbf{x}}$, $\bar{\mathbf{x}}_{aux}$, $\hat{\mathbf{x}}$

19   **Loss:**

$$\mathcal{L} = \left[ \|\mathbf{x} - \bar{\mathbf{x}}\|_2^2 + \lambda \cdot \|\hat{\mathbf{x}} - \bar{\mathbf{x}}_{\text{aux}}\|_2^2 \right]$$

---

For clarity, we describe our proposed method, multi-step sampling with progressive latent calibration, as pseudo-code in Algorithm 1.

**Generalization on different texture types.** In the 4th row of Fig. 8, the same 3D objects with different textures are illustrated with 3d objects, *motorcycle*. As shown, our method operates consistently across various textures and surface styles such as amigurumi and origami motorcycles. The high-quality results from these diverse texture prompts confirm that PLC works consistently to generate a wide range of detailed textures effectively.

## C    DIVERSITY EVALUATION

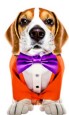 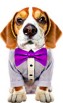 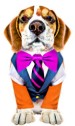          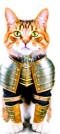 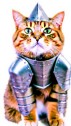 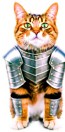

"A beagle in a detective outfit"                    "a DSLR photo of a cat wearing armor"

Figure 7: **Diversity evaluation results.** The samples are generated from two prompts, *"A beagle in a detective outfit"* and *"A DSLR photo of a cat wearing armor"*, with different random seeds.

To verify the diversity of assets generated by our method, we visualized three samples per prompt for two prompts, *"A beagle in a detective outfit"* and *"A DSLR photo of a cat wearing armor"*, by varying the random seed. The results in Fig. 7 illustrate that different seeds yield distinct variations in outfit and armor styles, demonstrating our method's ability to produce a wide range of plausible and creative assets from the same prompt.

## D    MORE QUALITATIVE RESULTS

We provide additional qualitative results in Fig. 8 to further demonstrate the effectiveness and robustness of our method. The accompanying figures showcase diverse examples across different prompts, highlighting our model's ability to handle challenging cases with high fidelity and consistency.

## E    FAILURE CASES

**Multi-view inconsistency.** Our proposed method generates pseudo-ground-truths aligned with the original rendering through multi-step sampling with progressive latent calibration, which significantly reduces superfluous details or unnecessary artifacts. As a result, the Janus problem, a multi-view inconsistency issue common to all score distillation-based methods, is greatly mitigated. However, as shown in Fig. 9, the Janus problem still occasionally occurs.

**Compositional prompt.** While our method generally performs well in capturing object-centric semantics, it occasionally fails to fully realize compositional prompts, especially under certain random seeds as shown in Fig. 10. We believe that the challenges observed in handling complex compositional prompts can be mitigated by leveraging stronger priors from recent advances in text-to-image diffusion models such as Stable Diffusion 3.0 and 3.5 (Esser et al., 2024). These models demonstrate a remarkable ability to faithfully interpret and synthesize intricate prompts involving multiple objects, actions, and spatial relationships. We leave this as future work.

## F    RELATED WORKS

**Text-to-3D generation.** Text-to-3D generation has gained significant momentum in recent years, driven by advances in both diffusion models and 3D representations. Early works such as Dream-Fields (Jain et al., 2022) and CLIP-Mesh (Mohammad Khalid et al., 2022) leveraged CLIP-based supervision to optimize implicit 3D representations from text prompts, but often struggled with limited geometry fidelity and weak supervision. A major breakthrough came with DreamFusion (Poole et al., 2023), which proposed distilling gradients from a pretrained 2D text-to-image diffusion model into a NeRF representation through Score Distillation Sampling (SDS). This enabled high-quality 3D

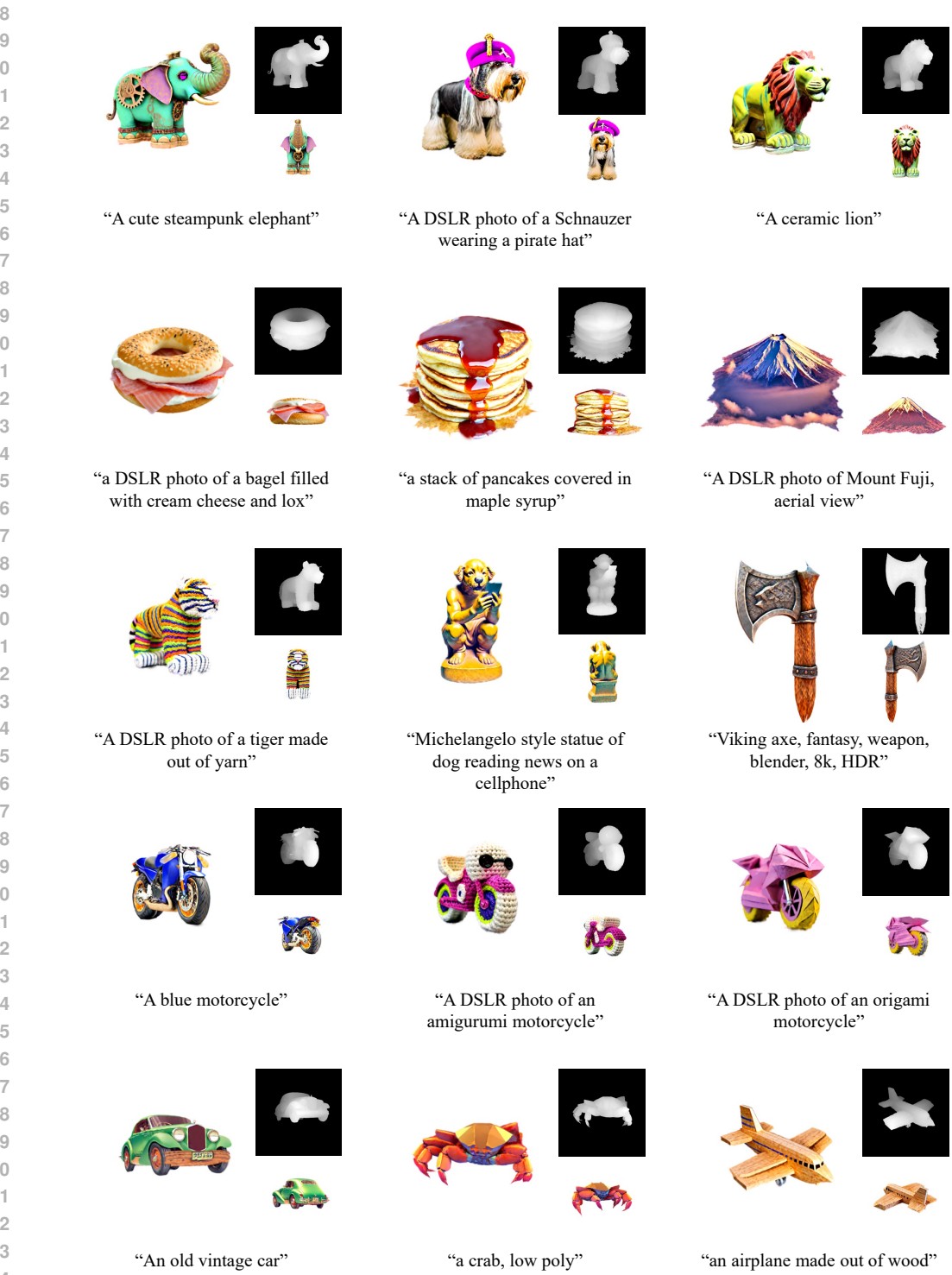

Figure 8: We present 15 additional assets generated by the proposed method. Each asset is shown together with its corresponding prompt, as listed below. All prompts are sourced from the DreamFusion Gallery (Poole et al., 2022).

asset generation from text alone, inspiring a series of follow-up works including Magic3D (Lin et al., 2023), Fantasia3D (Chen et al., 2023), and Text2Mesh (Michel et al., 2022), which extended the idea

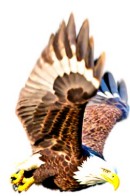 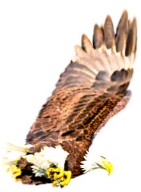 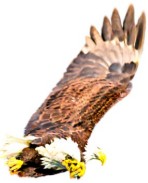 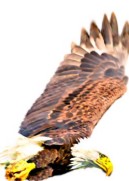

Figure 9: **A failure case from the proposed method exhibiting the Janus problem.** Given the prompt *"A DSLR photo of a bald eagle"*, the generated asset exhibits a failure where the eagle's claws, visible in the frontal view, incorrectly appear as a second head in the side view.

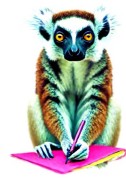 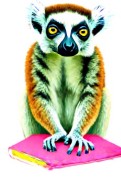 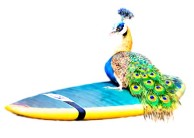 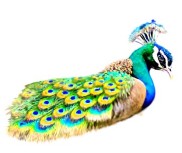

"A lemur taking notes in a journal"                "A DSLR photo of a peacock on a surfboard"

Figure 10: **Variation in prompt fidelity under different random seeds for compositional prompts.** The assets are generated using the proposed method with prompts *"A lemur taking notes in a journal"* and *"A DSLR photo of a peacock on a surfboard"*. For each prompt, samples on the left successfully capture the intended compositions, clearly depicting the specified actions or scenes. In contrast, samples on the right fail to fully reflect key elements of the prompts, such as "taking notes" and "surfboard", illustrating the challenges in faithfully rendering complex compositional instructions across random seeds.

to mesh-based pipelines or improved optimization stability and realism. The approaches (Poole et al., 2023; Wang et al., 2023a;b; Katzir et al., 2024; McAllister et al., 2024; Yu et al., 2024; Liang et al., 2024; Lukoianov et al., 2024; Li et al., 2024c; Zhuo et al., 2024; Wei et al., 2024) based on SDS highlighted the effectiveness of leveraging 2D diffusion priors for guiding 3D generation without explicit 3D supervision. Recent work has explored replacing diffusion-based priors with rectified flow models for text-to-3D generation. FlowDreamer Li et al. (2024a) uses a pretrained text-to-image rectified flow model and designs a trajectory-aware objective that improves 3D fidelity and convergence over SDS-style baselines. In parallel, RFDS Yang et al. (2025) defines rectified-flow analogues of SDS and shows that such priors benefit text-to-3D, inversion, and editing. Complementary to these methods, a growing number of approaches (Liu et al., 2023; Long et al., 2024; Shi et al., 2023; 2024; Wang et al., 2024b; Huang et al., 2024; Tseng et al., 2023; Li et al., 2024b) have adopted multi-view-consistent image generation as an intermediate step toward 3D synthesis. Models such as Zero123 (Liu et al., 2023) and Wonder3D (Long et al., 2024) generate novel views of a scene conditioned on a single or few reference images and a text prompt, enforcing geometric coherence across views. These synthetic multi-view images are then used for 3D reconstruction, often via NeRF or Gaussian Splatting. This direction avoids gradient-based 3D optimization entirely and benefits from the scalability and compositional priors of pretrained diffusion models. Together, these developments reflect a broader trend in text-to-3D research: bridging the gap between powerful 2D generative models and efficient, high-fidelity 3D representations through novel optimization strategies and view-consistent priors.

**Differentiable 3D representations** The field of 3D scene representation and rendering has witnessed significant progress through the development of neural rendering methods. Among them, Neural Radiance Fields (NeRF) (Mildenhall et al., 2020) has emerged as a foundational work, introduc-

ing a fully-connected neural network that models volumetric scenes by learning a mapping from spatial coordinates and viewing directions to color and density. NeRF enables photorealistic novel view synthesis from sparse input views, but suffers from slow training and inference due to its dense ray sampling and per-query MLP evaluations. To address the computational inefficiencies of NeRF, subsequent works have explored acceleration strategies. One notable advancement is Instant Neural Graphics Primitives (InstantNGP) (Müller et al., 2022), which replaces the MLP with a hash-encoded multi-resolution grid and trains with mixed-precision and CUDA-level optimizations. InstantNGP achieves real-time training and inference speeds, making neural rendering practical for interactive applications, while maintaining high visual fidelity. More recently, 3D Gaussian Splatting (3DGS) (Kerbl et al., 2023) has introduced a paradigm shift by moving away from implicit neural representations to an explicit point-based approach. Instead of volumetric or voxelized fields, 3DGS represents scenes as a collection of anisotropic Gaussians, each carrying spatial extent, color, and opacity information. This method leverages efficient rasterization pipelines and achieves state-of-the-art quality and rendering speed without requiring a learned radiance field, thereby dramatically reducing training time and improving scalability.

## G   MEMORY CONSUMPTION AND COMPUTATIONAL EFFICIENCY OF PLC

Table 4: Comparison of memory and runtime efficiency across different score distillation methods using RTX4090.

| Method | VRAM (MB) $\downarrow$ | Time (min) $\downarrow$ |
|--------|--------|--------|
| ISM | 21808 | 128 |
| GCS | 21555 | 168 |
| Ours | **21378** | 133 |

We compare the memory consumption and runtime for 3D asset generation with state-of-the-art score distillation methods in Fig. 4. The primary VRAM bottleneck in such frameworks arises from the 3D representation itself (in our case, Gaussian Splatting), as the diffusion models are used solely for inference rather than training. Consequently, the VRAM usage remains similar across all methods. In terms of training time, although our method adopts a multi-step strategy, it avoids the computationally expensive DDIM inversion process required by methods such as ISM and GCS. This makes our overall runtime highly competitive and comparable to other state-of-the-art approaches, demonstrating that our method achieves superior quality without incurring severe additional computational cost.

## H   EFFECTIVENESS OF THE PROPOSED OBJECTIVES

Table 5: Ablation results on the effect of the proposed losses.

| Method | CLIP Score $\uparrow$ | FID $\downarrow$ |
|--------|--------|--------|
| Multi-step sampling | 26.41 | 178.61 |
| + $\mathcal{L}_m$ | 32.63 | 108.80 |
| + $\mathcal{L}_{aux}$ | **32.68** | **100.41** |

To substantiate the impact of our proposed auxiliary loss $\mathcal{L}_{aux}$ in Fig. 7, we have conducted an ablation experiment in Fig. 5. The results show the contribution of each component. Our main contribution, the matching loss ($\mathcal{L}_m$), addresses the high variance problem in multi-step sampling, enabling the generation of a coherent object. On top of this, the auxiliary loss ($\mathcal{L}_{aux}$) provides further improvements by helping to steer the optimization direction toward the target prompt in the initial steps. This leads to enhanced performance in both CLIP Score and FID, confirming its effectiveness. Fig. 11 visualizes the 3d objects generated with multi-step sampling without PLC. With a moderate CFG scale (e.g., 7.5), naïve multi-step sampling for score distillation produces an over-smoothing artifact, in addition to the color saturation.

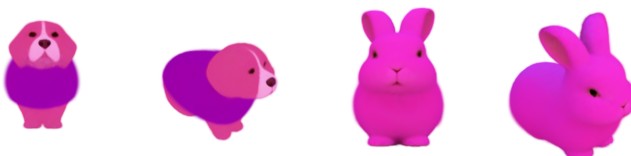

Figure 11: Visualization of the 3D object generated with multi-step sampling without PLC. Without explicit control over the stochasticity of random noise, multi-step sampling with moderate CFG scale (e.g., 7.5) suffers from the over-smoothing artifact in addition to the color saturation.

## I  ADDITIONAL INTUITION FOR PLC: A GUIDANCE PERSPECTIVE

In this section, we provide a further theoretical viewpoint that explains the PLC from the sampling guidance perspective. Specifically, PLC can be more analytically interpreted as a specific formulation of sampling guidance Ho & Salimans (2022) that pushes the sampling trajectory toward preserving the original semantics of $x_0$.

The sampling guidance of diffusion models can be generalized as the gradient of the energy function, where CFG is a specific case that guides samples toward enhancing a given class or prompt Ho & Salimans (2022); Ahn et al. (2024). Similarly, our residual term ($\epsilon_n - \epsilon_\phi(x_{s_n}; s_n, \emptyset)$ in Fig. 5) can be interpreted as the gradient of an implicit energy function, anchored to a fixed-camera rendering $x_{s_n}$. We regard the multi-step sampling process from the original rendering to the pseudo-GT as a generative function, based on which we design our energy function.

Let an energy function that discriminates the desirable distribution $p_d(y|x_{s_n})$ and the undesirable distribution $p_u(\hat{y}|x_{s_n})$, where $y$ and $\hat{y}$ are imaginary labels indicating whether the semantic fidelity to the original rendering is preserved or not, respectively. Then, similar to WGAN (Arjovsky et al., 2017), we can set the generator loss of the implicit discriminator as our energy function as $\mathcal{E}(x_{s_n}) = -\log \frac{p_d}{p_u}$. Given access to the $x_0$, we can obtain the GT noise $\epsilon$ corresponding to the sample that preserves semantic fidelity. Meanwhile, the score for the sample whose semantic feature is replaced by the diffusion prior can be approximated using diffusion noise prediction. Unlike the score $\hat{\epsilon}_\phi(\tilde{x}_{s_n,n}; s_n, y)$ in Fig. 5, which aims to predict high-fidelity noise based on $\tilde{x}_{s_n,n}$, $p_u$ is an undesirable distribution based on the rendering of current 3d object $x_{s_n,n}$. As a result, to approximate this, we utilize diffusion noise prediction with a null prompt $\epsilon_\phi(x_{s_n}; s_n, \emptyset)$, as this typically produces lower-quality outputs than conditional prediction, similar to aligned bad output in guidance literature (Karras et al., 2024; Hyung et al., 2025).

Based on this, the gradient of the energy function can be approximated as $\nabla x_{s_n} \mathcal{E} = \epsilon_\phi(x_{s_n}; s_n, \emptyset) - \epsilon_n$ similar to the derivation in Ahn et al. (2024). Injecting $-\nabla_{x_{s_n}} \mathcal{E}$ into the DDIM score, therefore, performs gradient descent on $\mathcal{E}$, progressively pulling the sampling trajectory toward the desirable manifold and repelling it from the unconditional prior. As a result, our corrected score injects high-frequency rendering information with reduced variance with a stable anchor $x_{s_n}$.

## J  GRADIENT CLIPPING CANNOT FULLY MITIGATE OVER-SATURATION

Table 6: Comparison of different sampling strategies with gradient clipping.

| Method | CLIP Score ↑ | FID ↓ | VRAM (MB) | Time (min) |
|---|---|---|---|---|
| Single-step sampling + grad clipping | 27.56 | 185.44 | 21265 | 75 |
| Multi-step sampling + grad clipping | 26.41 | 178.61 | 21329 | 112 |
| Multi-step sampling + grad clipping + PLC | **32.68** | **100.41** | 21378 | 133 |

Following the previous work (Li et al., 2024c), we incorporate gradient clipping (Pan et al., 2023) alongside our PLC framework. While gradient clipping alleviates the over-saturation issue of SDS, it alone fails to generate high-fidelity 3d assets. Tab. 6 shows the quantitative results with gradient

clipping with and without PLC. Applying only gradient clipping (in both single-step and multi-step settings) helps mitigate color oversaturation, but a severe oversmoothing effect remains, losing detailed textures. This is reflected in the poor CLIP and FID scores and visualized in Fig. 11. The table also shows that our PLC method achieves its substantial quality gains with only a minor increase in computational cost. Compared to the multi-step baseline, our method adds a negligible amount of VRAM usage. While the training time increases slightly, this modest overhead yields a massive leap in performance, as seen in the dramatic improvement of both CLIP and FID scores.

In conclusion, while gradient clipping addresses color oversaturation, it is not sufficient for generating high-fidelity details. Our Progressive Latent Calibration is the crucial component that resolves the high variance problem during optimization, enabling the generation of fine structures with minimal additional computational cost.

## K  IMPACT OF LIMITED INVERSION AND SAMPLING STEPS

Table 7: Reconstruction error after 30 iterative DDIM inversion–reconstruction.

| $N$ (Inversion) / $M$ (Sampling) | MSE | LPIPS |
|---|---|---|
| 50 / 50 | 0.208 | 0.678 |
| 50 / 10 | 0.209 | 0.729 |
| 10 / 50 | 0.300 | 0.807 |

As noted in Fig. 3.1, the limited number of DDIM inversions can introduce non-negligible error accumulation during reconstruction, often resulting in artifacts as shown in Fig. 2. To further clarify this, we quantitatively measure the reconstruction error in Fig. 7. Specifically, we recursively apply $N$-step DDIM inversion to the input image, followed by $M$-step DDIM sampling for reconstruction using Stable Diffusion 2.1. After 30 inversion–reconstruction, we measure MSE and LPIPS between the input and reconstructed images to evaluate accumulated errors. The results show that fewer inversion or sampling steps substantially increase reconstruction error.

## L  DETAILED ABLATION ON NUMBER OF PSEUDO-GT SAMPLING STEPS ($N$)

To justify our choice of the number of sampling steps ($N$) in the multi-step pseudo-ground-truth sampling process, we conducted an ablation study evaluating the trade-off between generative quality (FID, CLIP Score) and computational cost (Runtime). The core finding of this ablation is that while cost increases linearly with $N$, the quality stabilizes quickly due to the diminishing returns of the calibration process.

The table 8 summarizes the performance metrics for varying step counts. The quantitative results demonstrate that the model exhibits insufficient performance when $N$ is smaller than 4 (i.e., $N = 3$), with notably worse FID and CLIP scores. This suggests that three steps are inadequate for generating a high-fidelity pseudo-GT. Conversely, increasing $N$ beyond 4 yields no significant changes in the quantitative metrics. While the runtime increases linearly with $N$, VRAM usage remains constant, as this process does not allocate additional model parameters.

Table 8: Quantitative Ablation on the Number of Sampling Steps ($N$)

| N (Steps) | FID $\downarrow$ | CLIP Score $\uparrow$ | Runtime (min) |
|---|---|---|---|
| 3 | 108.15 | 32.59 | 104 |
| 4 (Ours) | 100.41 | 32.68 | 133 |
| 5 | 102.08 | 32.69 | 150 |
| 6 | 101.63 | 32.81 | 160 |

Figure 12 visually confirms the necessity of adequate sampling steps. At $N = 3$, the assets (e.g., more blurred yarn, details on the Schnauzer's beard/jaw, hat, and eyes) lack the intricate texture and

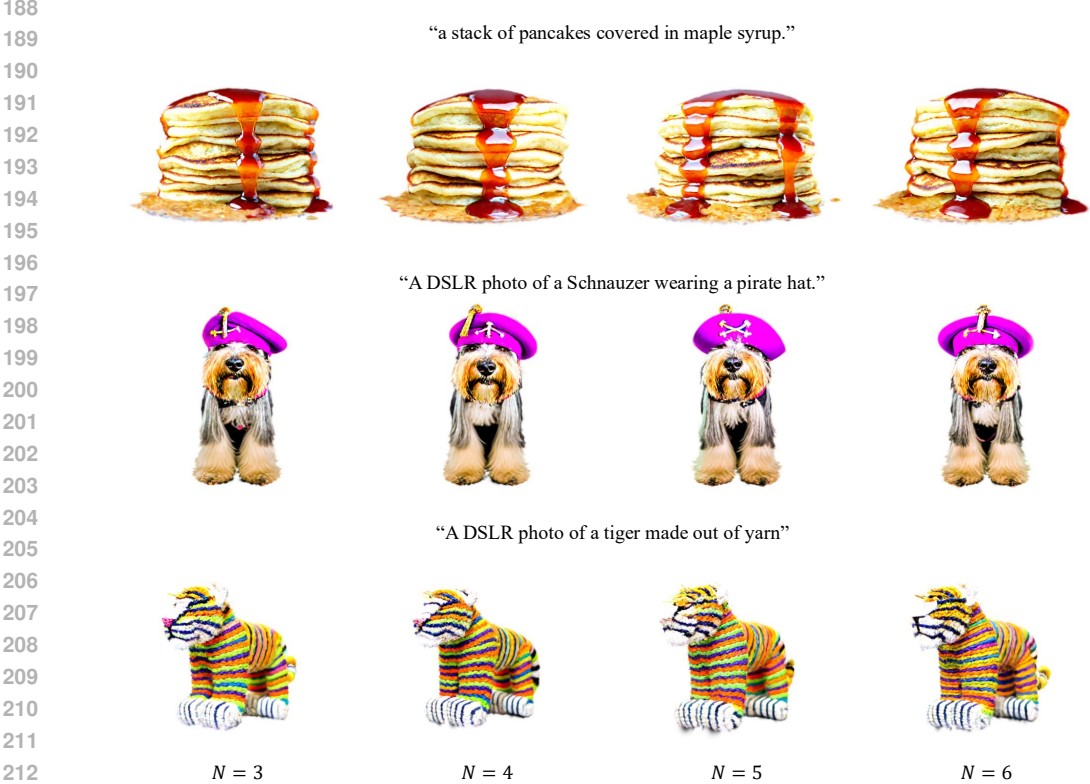

"a stack of pancakes covered in maple syrup."

"A DSLR photo of a Schnauzer wearing a pirate hat."

"A DSLR photo of a tiger made out of yarn"

$N = 3$      $N = 4$      $N = 5$      $N = 6$

Figure 12: Qualitative comparison across different number of sampling steps ($N$) for PLC.

semantic resolution present at $N = 4$. As $N$ increases from $4$ to $6$, the visual quality stabilizes, and the performance difference becomes minor, primarily restricted to subtle variations in texture sharpness. Given the minimal additional gain beyond $N = 4$ versus the linear increase in computational cost, we confirm that $N = 4$ provides the optimal trade-off between generation fidelity and efficiency for our PLC framework.

## M    ABLATION STUDY ON THE RESIDUAL SCALING FACTOR

We conducted a dedicated qualitative ablation study on the scaling factor ($s_r$) applied to the latent calibration residual, formulated as $s_r \times (\epsilon_n - \epsilon_\phi(x_{s_n}; s_n, \emptyset))$. This factor controls the magnitude of the noise residual injected back into the diffusion sampling path. We evaluated a range of scales ($s_r \in \{0.5, 0.8, 1.0, 1.2, 2.0\}$) to determine the optimal balance between detail preservation and stability. As shown in Fig. 13, the choice of $s_r$ significantly impacts generation quality:

- **Weak correction ($s_r = 0.5$):** The corrective signal is too weak to counteract the diffusion model's tendency to average out details. Consequently, the result suffers from severe over-smoothing, failing to recover fine textures or accurate colors, showing similar artifacts in the results without PLC (fig. 11).

- **Near-optimal scales ($s_r = 0.8, 1.2$):** Values close to unity yield structurally coherent results but exhibit distinct imperfections. At $s_r = 0.8$, while the overall shape is improved compared to $0.5$, visible over-smoothing artifacts remain, and the image lacks the sharp definition seen at the optimal scale. Conversely, increasing the scale slightly to $s_r = 1.2$ begins to introduce color saturation artifacts, which are particularly noticeable on the astronaut's backpack. This indicates that the latent trajectory is starting to deviate from the natural image manifold.

"An astronaut is riding a horse"

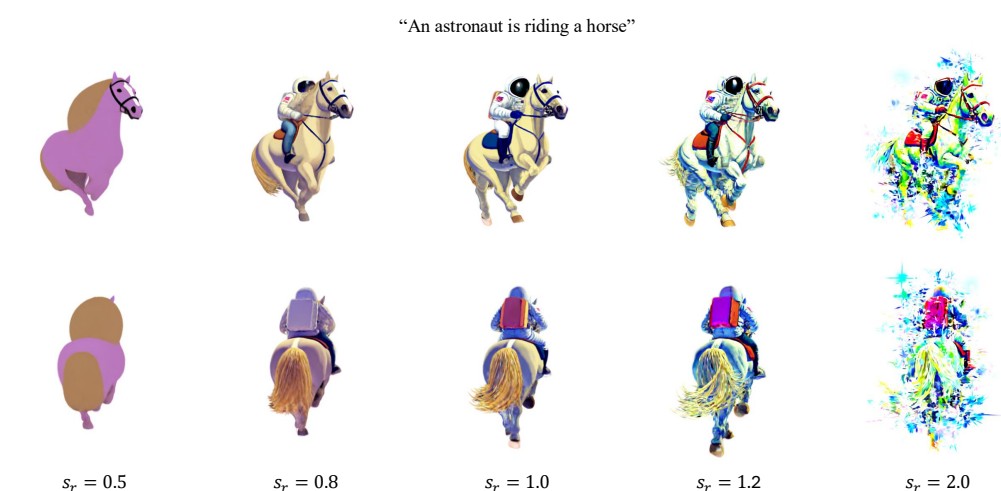

| $s_r = 0.5$ | $s_r = 0.8$ | $s_r = 1.0$ | $s_r = 1.2$ | $s_r = 2.0$ |

Figure 13: Qualitative analysis of residual scaling factor ($s_r$)

- **Over-correction** ($s_r = 2.0$): The saturation artifacts observed at 1.2 escalate into severe divergence. The calibration term acts as a strong extrapolation force, pushing the sampling trajectory completely off-manifold. This results in chaotic noise injection and structural breakdown.

These observations show that using values of the scale parameter $s_r$ that differ substantially from $s_r = 1.0$ can harm the fidelity of the generated samples. This behavior is consistent with prior work. In work Hong et al. (2023), which uses perturbed inputs as guidance for diffusion models, small scaling factors are used to avoid pushing the latent state into off-manifold regions. Similar to these results, we set $s_r = 1.0$ in all experiments, which provides stable and effective calibration and maintains a good balance between detail preservation and manifold stability.

## N  APPLYING PLC ON SDEDIT FOR FAITHFUL IMAGE TRANSLATION

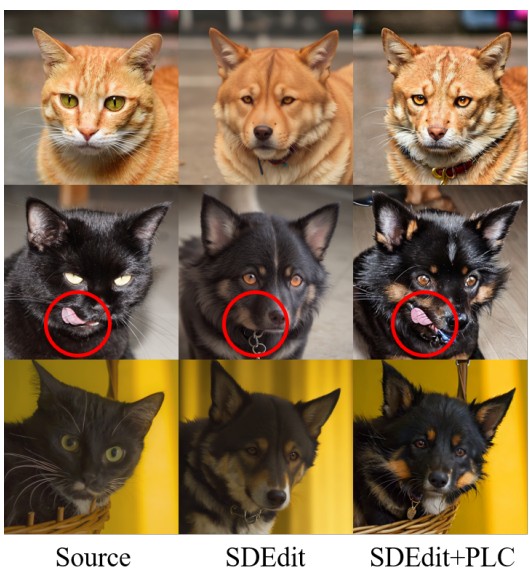

Source          SDEdit          SDEdit+PLC

Figure 14: Applying PLC on Cat→Dog I2I translation with SDEdit (Meng et al., 2021). PLC can effectively enhance the faithfulness of the output by preserving source information.

Table 9: Quantitative results on Cat→Dog I2I translation.

| Method | FID ↓ | LPIPS ↓ | SSIM ↑ | PSNR ↑ |
|---|---|---|---|---|
| SDEdit | 113.75 | 0.544 | 0.405 | 15.70 |
| SDEdit + PLC | **99.67** | **0.435** | **0.450** | **16.18** |

The residual term in PLC preserves source information that would otherwise be lost during random perturbations and repetitive denoising. This property is particularly useful in the image-to-image (I2I) translation task, where maintaining the semantic information of the source image is essential. I2I performance is typically assessed along two dimensions: (i) realism, i.e., whether the source image is convincingly translated into the target domain (or aligns with the target prompt), and (ii) faithfulness, i.e., whether the structural and semantic properties of the source image are preserved.

As a preliminary study, we integrate PLC into SDEdit (Meng et al., 2021) to enhance the faithfulness of I2I translation. We conduct a Cat→Dog translation experiment using the AFHQ dataset (Choi et al., 2020). The input images are perturbed up to $t = 0.5T$ where we set $T = 20$, and denoised using the LCM model Luo et al. (2023) for efficiency. Following standard I2I evaluation protocols, we report FID for realism and LPIPS, SSIM, and PSNR for faithfulness. The results in Fig. 9 show that the proposed PLC effectively preserves the source identity, without harming the realism and generative performance. Qualitative comparison on Fig. 14 clearly shows that PLC effectively preserves the source information, such as the tongue in the second row and the basket in the third row, which can be dropped by random perturbation and SDEdit fails to preserve.

## O    EXTENDED QUALITATIVE COMPARISONS

As discussed in the main manuscript, the 3D object generated by our proposed method appear significantly more realistic, whereas previous methods often exhibit unnatural color tones. Moreover, in the zoomed-in views in Fig. 15, we observe that all baseline methods produce noticeable artifacts resembling black blotches, while our method preserves clean object boundaries without such distortions.

## P    USER PREFERENCE TEST INTERFACE

To conduct the user preference test efficiently and remotely, we implemented a lightweight web-based evaluation interface using Streamlit (Inc., 2025b), a Python framework for building interactive applications. The interface presented participants with three rendered 3D asset videos for each prompt and asked them to select their preferred result for three criteria (overall quality, prompt alignment, and color fidelity). To share the interface publicly without deploying to a cloud server, we used ngrok (Inc., 2025a) to expose the local Streamlit application via a secure public URL. This allowed participants to access the test interface from any device without requiring additional installation or configuration. An example screenshot of the user study interface is shown in Fig. 16, demonstrating the layout of the video comparisons and the associated multiple-choice questionnaire.

## Q    ANONYMOUS PROJECT PAGE

In the abstract of our main manuscript, we provide a link to the anonymous project page for this paper. We post videos of the rendered 3D assets included in the experiments of the main manuscript on the project page. We have constructed an anonymized project page in adherence to the anonymity policy and last updated this project page on September 24, 2025, at 08:00 PM (AoE), before the deadline of the main manuscript.

## R    THE USAGE OF LARGE LANGUAGE MODELS

We employed large language models (LLMs) as a general-purpose assistive tool to improve the presentation quality and efficiency of this work. Specifically, the LLM was used for refining phrasing,

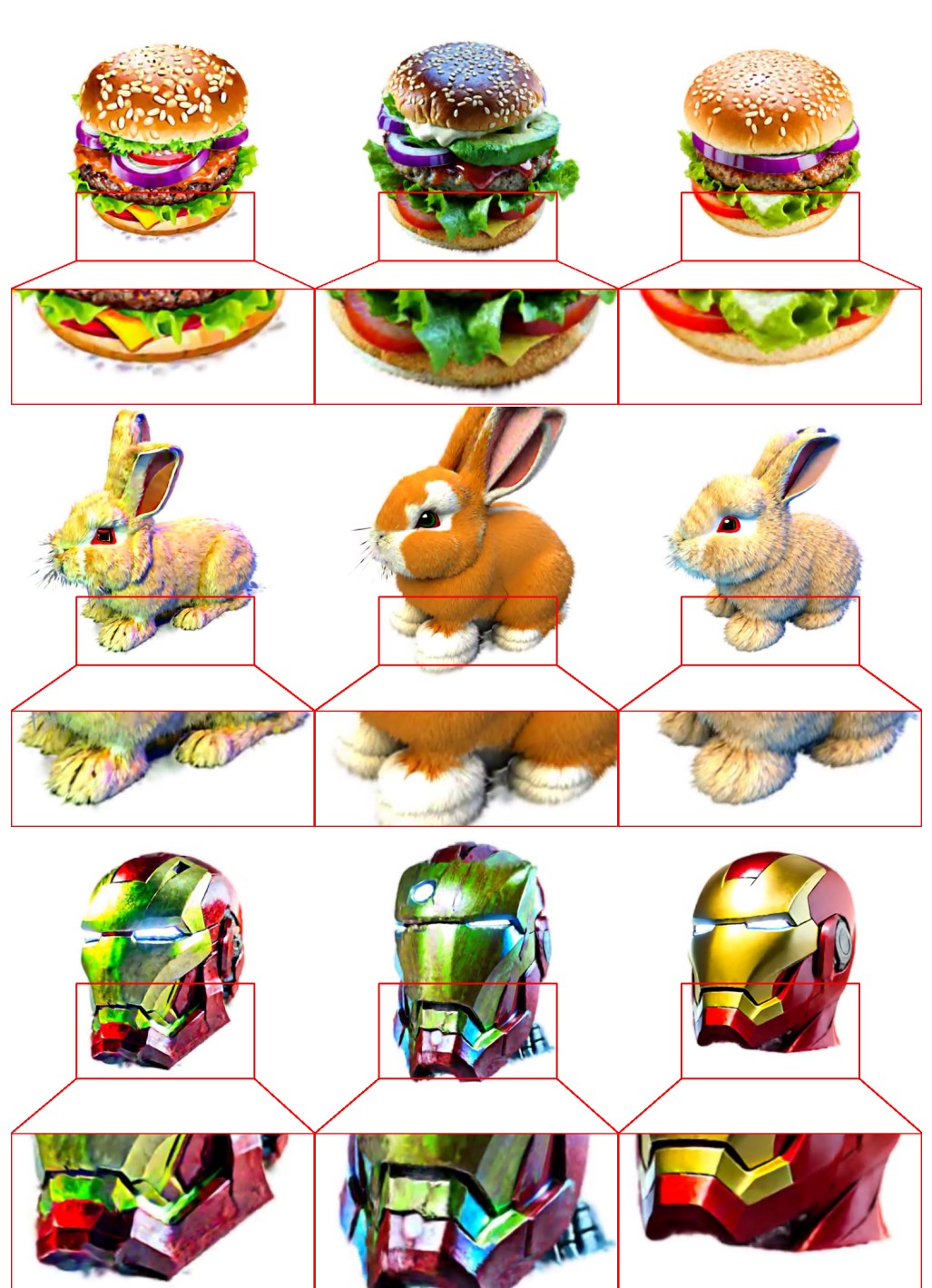

Figure 15: **Detailed results for closer inspection.** From left to right, the results correspond to ISM (Liang et al., 2024), GCS (Li et al., 2024c), and the proposed method.

proofreading for grammatical and typographical errors, assisting with LaTeX formatting, and offering support with coding tasks such as debugging and syntax adjustments. The LLM was not involved in the generation of research ideas, experimental design, or data analysis.

# 3D Asset User Preference Study

> **Before answering each question, please make sure to watch all 3 videos completely.**
>
> **These videos are critical for evaluating 3D asset quality. Skipping them may distort the results.**
>
> **The order of videos is randomized for each prompt.**

## Prompt 1: 'A photo of the IRONMAN, highly realistic DSLR photo'

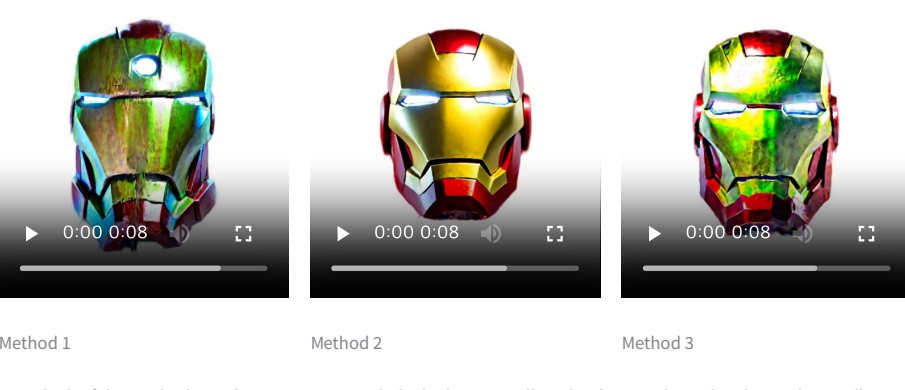

Method 1      Method 2      Method 3

Q1: Which of the methods produces 3D assets with the highest overall quality (e.g., realism, detail, visual appeal)?

- ◉ Method 1
- ○ Method 2
- ○ Method 3

Q2: Which of the following methods aligns most with the text prompt?

- ◉ Method 1
- ○ Method 2
- ○ Method 3

Q3: Which of the methods produces 3D assets with the most natural and realistic color appearance?

- ◉ Method 1
- ○ Method 2
- ○ Method 3

Figure 16: **Web interface used for the user preference study.** Participants were shown three 360° rendered videos side-by-side for each prompt. For each prompt, users selected their preferred result based on (1) overall quality, (2) prompt alignment, and (3) naturalness of color appearance. The interface was implemented using Streamlit and shared via a secure public link using ngrok.

