# OpenReview forum: "Progressive Latent Calibration for Stable Score Distillation"
_ICLR.cc/2026/Conference — Submitted to ICLR 2026_

### Official Review · Reviewer_LKaP · 2025-10-28

**Soundness:** 3
**Presentation:** 3
**Contribution:** 3
**Rating:** 6
**Confidence:** 4

**Summary:**

This paper introduces Progressive Latent Calibration (PLC), a novel score distillation framework for text-to-3D generation designed to overcome critical limitations in existing methods. The authors first identify that while standard Score Distillation Sampling (SDS) produces over-smoothed results due to high variance, recent inversion-based methods suffer from the accumulation of discretization errors from repeated DDIM inversion, leading to significant artifacts like color degradation and structural distortion. To solve this, PLC proposes an inversion-free strategy that utilizes multi-step pseudo-GT sampling. The core mechanism "progressively calibrates" this sampling process by explicitly estimating and reintegrating information loss about the original 3D rendering at each step; this is achieved by adding a corrective noise residual which is the difference between the actual noise and an unconditional prediction, back into the guided denoising step. This method successfully reduces variance and preserves high-frequency details without relying on DDIM inversion, demonstrably outperforming existing approaches in generating high-fidelity 3D assets.

**Strengths:**

1.	The paper clearly identifies and diagnoses a critical, yet previously overlooked, problem in score distillation: the accumulation of discretization errors from repeated DDIM inversion, which leads to artifacts like color degradation.
2.	The proposed Progressive Latent Calibration (PLC) framework is a highly novel inversion-free alternative.
3.	extensive and high-quality evaluations, including thorough quantitative comparisons against a strong set of baselines, compelling qualitative results, and a user preference study that validates the perceptual quality of the generated 3D assets .

**Weaknesses:**

1.	The proposed PLC method remains essentially a 2D operation focused on improving single-view fidelity, without introducing priors or constraints for multi-view consistency. As acknowledged, it still suffers from the Janus problem.
2.	The cost analysis is misleading—PLC scales linearly with sampling steps and requires multiple U-Net passes, making it substantially more expensive than single-step baselines.
3.	The introduced auxiliary loss $\mathcal{L}_{aux}$ adds complexity but offers marginal benefit. The paper’s own ablations show that most performance gains stem from the main PLC-guided loss $\mathcal{L}m$, while $\mathcal{L}{aux}$ yields negligible quantitative and qualitative improvements.

**Questions:**

The paper could be improved by comparing with flow matching based method like FLowDreamer[1] or RFDS[2], which addresses trajectory inconsistency via Rectified Flow rather than stochastic heuristics like PLC, offering a deterministic and potentially cleaner alternative.
[1]. Li H, Chu X, Shi D, et al. Flowdreamer: Exploring high fidelity text-to-3d generation via rectified flow[J]. arXiv preprint arXiv:2408.05008, 2024.
[2]. Yang X, Chen C, Yang X, et al. Text-to-image rectified flow as plug-and-play priors[J]. arXiv preprint arXiv:2406.03293, 2024.

---

> ### Author Response · Authors · 2025-11-21
>
> We sincerely thank Reviewer LKaP for the detailed and accurate summary of our work, and for recognizing the novelty of our problem diagnosis (accumulation of discretization errors from repeated DDIM inversion) and our inversion-free PLC framework.
>
> We address the weaknesses and questions below.
>
> ---
>
> ### **Weakness 1: Multi-View Consistency**
>
> We agree that multi-view consistency is a critical challenge in text-to-3D generation. The primary scope of our work was to solve the high-variance problem inherent in the score distillation sampling process, so we did not directly focus on multi-view consistency as our main contribution. However, we observed that severe consistency issues were not as prominent in our implementation because we initialized our scenes with the Point-E model following the ISM setup, which provides a strong geometric prior, and also applied Perp-Neg to effectively resolve the Janus problem.
>
> To more directly address multi-view consistency, a key advantage of our method is its plug-and-play nature. Our Progressive Latent Calibration (PLC) can be easily integrated with other works that specifically target this challenge, such as Collaborative Score Distillation [1] or Geometry-Aware Score Distillation [2, 3]. We are confident that combining these approaches would be an effective way to address both the high-variance and multi-view consistency problems simultaneously. Unfortunately, due to the limited rebuttal period, we were unable to conduct these integration experiments, but we consider it a very promising direction for future work.
>
> [1] Kim, Subin, et al. "Collaborative score distillation for consistent visual synthesis.", NeurIPS 2023.
>
> [2] Seo, Junyoung, et al. "Let 2D Diffusion Model Know 3D-Consistency for Robust Text-to-3D Generation.", ICLR 2024.
>
> [3] Kwak, Min-Seop, et al. "Geometry-Aware Score Distillation via 3D Consistent Noising and Gradient Consistency Modeling.", arXiv 2024.
>
> ---
>
> ### **Weakness 2: Cost Analysis**
>
> As the reviewer correctly pointed out, PLC is indeed more computationally expensive than single-step baselines such as standard SDS. However, as discussed in prior literature, single-step methods inherently suffer from severe over-smoothing and produce significantly lower-quality assets. This low performance is consistently reflected in their poor quantitative scores.
>
> Our claim is that PLC is highly competitive when compared to other high-fidelity methods. As shown in Table 1, despite using multi-step sampling, PLC's runtime (133 min) is comparable to ISM (128 min), faster than GCS (168 min), and significantly faster than VSD (254 min).
>
> ---
>
> ### **Weakness 3: The introduced auxiliary loss** $\mathcal{L}_{aux}$ **adds complexity but offers marginal benefit.**
>
> Although the mathematical formulation of the auxiliary loss may appear complex, it is straightforward to implement the loss in practice. It is defined directly on the same intermediate predictions that are already computed for the main matching loss, and therefore does not require any additional diffusion U-Net forward passes or new network components, as stated in the original manuscript (L311-322).  In practice, this leads to virtually no implementation complexity and only a negligible difference in computational cost, while consistently improving prompt alignment and class-discriminative details in our ablations. For this reason, we view the auxiliary term as a lightweight but practically useful addition to PLC rather than a source of meaningful overhead.
>
> ---
>
> ### **Question 1: Comparing with the Flow Matching-Based method**
>
> Following the reviewer’s suggestion, we have added a description of FlowDreamer and RFDS in the Related Works section (Appendix F) of the revised manuscript and compared their scope and design choices with those of PLC.

---

### Official Review · Reviewer_FiWq · 2025-10-30

**Soundness:** 4
**Presentation:** 3
**Contribution:** 3
**Rating:** 6
**Confidence:** 3

**Summary:**

This paper uses a score distillation approach to generate 3D models guided by text prompts.  In this framework, images of the 3D model are rendered and diffusion methods are used to guide the evolution of the 3D model by determining how to adjust the model to match denoised images.  The paper identifies a potential issue with current approaches and introduces a technique that adds a corrective noise during this iterative process to produce a more accurate trajectory.  Quantitative and qualitative experiments support the potential gains of this method.

**Strengths:**

The proposed method seems to be well motivated by solid intuitions.

Experiments indicate significant gains in performance.

**Weaknesses:**

The proposed method is somewhat incremental to existing approaches.  Results relative to ISM seem better, but not by a lot.

The paper is not really self-contained, and assumes implicitly some knowledge of existing approaches.  This would be hard to avoid, as the method builds on fairly complex prior work.

**Questions:**

In the user preference test, why was the method not compared to VSD?

Is it possible to measure how much of the gain from the method is due to better color, and how much is due to other improvements?

---

> ### Author Response · Authors · 2025-11-21
>
> We sincerely thank Reviewer FiWq for the constructive review, for recognizing that our method is well motivated by solid intuitions, and for noting that our experiments indicate significant gains.
>
> We address the weaknesses and questions below.
>
> ---
>
> ### **Weakness 1: Incremental Performance**
>
> We believe our contribution offers a substantial performance improvement, particularly by resolving the critical issues of color degradation and artifacts that persist in current SOTA methods. As evidenced in Figure 4 of the main manuscript and Figure 15 in Appendix O of the revised manuscript, our method produces significantly more natural colors and effectively reduces visual artifacts compared to existing baselines. These visual improvements directly address the severe over-saturation and color degradation problems that plague current state-of-the-art inversion-based methods. While we have already demonstrated superior performance in standard metrics like FID and CLIP score, we agree with the reviewer that a specific quantitative evaluation was needed to fully substantiate our claims regarding color fidelity. To address this, we have additionally measured saturation and contrast scores. These new metrics quantitatively validate that our method successfully resolves color degradation, achieving a balance closest to real images. Detailed results and discussions are provided in the Revised Manuscript (Section 4.5) and our response to Question 2. We believe that a review of these comprehensive qualitative and quantitative results will demonstrate that our method offers a significant performance leap, solving fundamental issues in score distillation that prior works could not.
>
> ---
>
> ### **Weakness 2: Self-Containedness**
>
> We agree that understanding our contribution benefits from familiarity with diffusion models, DDIM inversion, score distillation, and diffusion guidance. For this reason, we devoted Section 2 (Background) and Appendix F (Related Works) to summarizing the preliminaries most relevant to our method, and we explicitly cited the original sources where a more detailed treatment is required. Due to the page limit, we were unable to reproduce full derivations or extensive tutorials on these topics, which may have contributed to the impression that the paper is not fully self-contained. Nevertheless, all components that are specific to our contribution, including PLC and its integration into the score distillation pipeline, are defined and motivated within the main text. We therefore expect that readers with a background in diffusion-based generative modeling and text-to-3D will be able to follow the proposed method after consulting the referenced background material.
>
> ---
>
> ### **Question 1: Rationale for Excluding VSD from User Preference Test**
>
> Our primary goal was to propose a stable, inversion-free score distillation framework. Therefore, our primary comparison targets for the user study were the SOTA inversion-based methods that our work directly aims to replace.
>
> Furthermore, VSD utilizes a different 3D representation compared to our method and the selected baselines, which makes it difficult to isolate the effect of the distillation method from that of the underlying 3D representation. VSD is implemented with an Instant-NGP representation, whereas our framework and our main baselines are built upon 3D Gaussian Splatting (3DGS). This difference in underlying 3D structure leads to fundamentally different results, poses, and appearances, as visually demonstrated in Figure 4 of the manuscript. Therefore, a direct preference test would make it impossible to determine if users were preferring the distillation method (PLC vs. VSD) or the underlying 3D representation (3DGS vs. Instant-NGP).
>
> Although VSD could be adapted to 3DGS to enable a direct comparison, this would demand substantial and non-trivial tuning of hyperparameters and configurations. Applying VSD to 3DGS without such extensive tuning would likely yield suboptimal results and an unfair assessment. Thus, to ensure a controlled and reliable comparison of distillation quality, we focused the user study exclusively on methods that are already natively optimized for the 3DGS representation.

---

> ### Author Response · Authors · 2025-11-21
>
> ### **Question 2: Quantitative Color Fidelity Measurement**
>
> Thank you for the valuable question that highlights the improvements in color perspective. To directly assess the color improvement, we measure the saturation score following Sadat et al. [1].
>
> Specifically, it first converts each image from RGB to HSV and computes the mean of the saturation channel. Then RMS contrast is calculated as the standard deviation of pixel values after converting the image to grayscale. The final metrics are derived by averaging the saturation and RMS contrast values across all images. We utilize the same images utilized for FID and CLIP score measuring in the main paper. We exclude the background for calculating metrics.
>
> The table below clearly shows that compared to GCS and ISM, our method shows closer saturation and contrast with real images and SD-generated images. While SDI shows a closer saturation score to real images, it shows a significantly lower contrast score compared to real images. This is due to severe color degradation as demonstrated in Fig. 4 of the paper.
>
> | **Method** | **Saturation** | **Contrast** |
> | --- | --- | --- |
> | COCO-val (Real Images) | 0.319 | 0.232 |
> | StableDiffusion Generated Samples | 0.336 | 0.234 |
> | PLC (Ours) | 0.439 | 0.221 |
> | GCS | 0.505 | 0.245 |
> | ISM | 0.522 | 0.248 |
> | SDI | 0.362 | 0.140 |
>
> We have included a new section (Section 4.5) detailing saturation score measurement in the revised manuscript.
>
> ---
>
> [1] Sadat, Seyedmorteza, Otmar Hilliges, and Romann M. Weber. "Eliminating oversaturation and artifacts of high guidance scales in diffusion models." ICLR 2025.

---

### Official Review · Reviewer_tCwW · 2025-11-01

**Soundness:** 3
**Presentation:** 3
**Contribution:** 2
**Rating:** 6
**Confidence:** 4

**Summary:**

Progressive Latent Calibration introduces an alternative to the Score Distillation Sampling (SDS) gradient in the Dreamfusion text-to-3D optimization framework. The method identifies problems with diffusion inversion (solving the PF-ODE forward in time) to obtain view-consistent latents and proposes to instead use the typical sampling-based diffusion forward process with a modified denoising process. This reduces the variance of the gradient, improving semantic and visual fidelity.

**Strengths:**

- Progressive latent calibration introduces a correction term in the denoising that pulls the denoising toward the current clean sample/render, an approach that has been effective in other settings
- Analysis on the deficits of diffusion inversion (particularly compounding errors) in this context is reasonable
- PLC is less computationally expensive than DDIM inversion because it doesn't require NFEs for the forward process
- User study is the most reliable way to evaluate such methods due to the lack of reliable metrics
- Results seem qualitatively good

**Weaknesses:**

- Latent calibration is mostly heuristically motivated, though it is reasonable
- Ablations do not probe sensitivity to many hyperparameters, for instance the residual scaling or the denoising schedule (step lengths)
- Methods in this problem setting are known to have inconsistent performance across text prompts or random seeds. It would be useful to see failure modes and how they compare to other methods.
- Scope of the method's contribution is relatively limited.

**Questions:**

- How well does it apply to other versions of Stable Diffusion, particularly SDXl, which is known to have problems with SDS

---

> ### Author Response · Authors · 2025-11-21
>
> We sincerely thank Reviewer tCwW for the insightful comments and for recognizing the value of our analysis on diffusion inversion deficits and the reliability of our user study and qualitative results.
>
> We address the weaknesses and questions below.
>
> ---
>
> ### **Weakness 1: Theoretical Perspective on Latent Calibration**
>
> While the motivation is intuitive, we provide a theoretical interpretation in Appendix I, where PLC is formulated as a specific case of sampling guidance. Specifically, the residual term acts as the gradient of an implicit energy function that anchors the diffusion trajectory to the original rendering, preventing it from drifting off-manifold.
>
> ---
>
> ### **Weakness 2: Ablations on Hyperparameters (Step Lengths & Residual Scaling)**
>
> Thank you for pointing out the need for these ablations. We have conducted ablation studies on both hyperparameters as requested.
>
> 1. Denoising Schedule (Step Lengths, $N$):
>
>     As detailed in our response to Reviewer J8ge, we performed an ablation on the number of sampling steps $N \in \{3, 4, 5, 6\}$. We found that $N=3$ resulted in a noticeable drop in quality (FID 108.15) compared to $N=4$ (FID 100.41), suggesting it is insufficient for generating a high-fidelity pseudo-GT. Increasing to $N=5$ (FID 102.08) or $N=6$ (FID 101.63) yielded only marginal differences within the margin of error, while incurring a linear increase in runtime. Therefore, we concluded that $N=4$ provides the optimal trade-off between generation quality and computational efficiency. We also included additional corresponding visual samples in the supplementary material (Figure 12 of Appendix L) of our revised manuscript to allow for a direct qualitative assessment of this.
>
> 2. Residual Scaling:
>
>     We also performed an ablation on the scaling factor for our progressive latent calibration term $(\epsilon_n - \epsilon_\phi(\dots, \emptyset))$, which is set to 1.0 in the paper. To visually confirm this finding, we have included additional qualitative samples (Figure 13 of Appendix M) in the supplementary material of our revised manuscript.
>
>     - Weak correction ($s_r = 0.5$): The corrective signal is too weak to counteract the diffusion model's tendency to average out details. Consequently, the result suffers from severe over-smoothing, failing to recover fine textures or accurate colors.
>     - Near-optimal scales ($s_r = 0.8, 1.2$): Values close to 1.0 maintain the overall structure, but they still show visible defects. At $s_r = 0.8$, while the overall shape is improved compared to 0.5, visible over-smoothing artifacts remain, and the image lacks the sharp definition seen at the optimal scale. Conversely, increasing the scale slightly to $s_r = 1.2$ begins to introduce color saturation artifacts, which are particularly noticeable on the astronaut's backpack. This indicates that the latent trajectory is starting to deviate from the natural image manifold.
>     - Over-correction ($s_r = 2.0$): The saturation artifacts observed at 1.2 escalate into severe divergence. The calibration term acts as a strong extrapolation force, pushing the sampling trajectory completely off-manifold. This results in chaotic noise injection and structural breakdown.
>
>     These observations show that using values of the scale parameter $s_r$ that differ substantially from $s_r = 1.0$ can harm the fidelity of the generated samples. This behavior is consistent with prior work. In the work [1], which uses perturbed inputs as guidance for diffusion models, small scaling factors are used to avoid pushing the latent state into off-manifold regions. Similar to these results, we set $s_r = 1.0$ in all experiments, which provides stable and effective calibration and maintains a good balance between detail preservation and manifold stability.
>
> The results and analyses of both experiments are also included in the revised manuscript. (Appendix L and M)
>
> [1] Hong, Susung, et al. "Improving sample quality of diffusion models using self-attention guidance." ICCV 2023.

---

> ### Author Response · Authors · 2025-11-21
>
> ### **Weakness 3: Analysis of Failure Modes**
>
> We acknowledge these limitations and have included a dedicated analysis in the supplementary material, Appendix E (Failure Cases) of our original manuscript. While these are different from the main problem we focus on in this work, we believe that addressing these limitations, which are common to score distillation approaches, would constitute a promising direction for future research.
>
> ---
>
> ### **Weakness 4: Scope of the Paper**
>
> Regarding the scope of our contribution, we respectfully wish to highlight that our work offers a significant advancement in the field by first systematically identifying a fundamental bottleneck: that accumulated discretization errors from repeated DDIM inversion are the root cause of severe artifacts, such as color degradation and structural distortion, in SOTA inversion-based methods. Building on this diagnosis, we establish a novel paradigm with Progressive Latent Calibration (PLC), an inversion-free framework that successfully bridges the gap between the efficiency of standard SDS and the high fidelity of expensive methods like VSD. Furthermore, as highlighted by our new quantitative analysis on color fidelity (Section 4.5 of the revised manuscript), PLC effectively resolves persistent usability issues regarding unrealistic colors, which we believe is a substantial step forward for the practical adoption of text-to-3D models.
>
> ### **Question 1: Applicability to SDXL**
>
> As the reviewer notes, and as confirmed by recent work [2], applying SDS to SDXL is a known challenge, widely attributed to the inherent instability of gradients originating from the SDXL VAE. The primary focus of our paper was to identify and solve a distinct but critical problem: the error accumulation from repeated DDIM inversion and the high variance of pseudo-GTs. Our proposed PLC framework is designed specifically to address this issue by stabilizing the diffusion sampling trajectory. While addressing VAE gradient instability is a separate research problem, since PLC is model-agnostic, it naturally complements techniques targeting VAE instability. Therefore, we anticipate that combining PLC with methods like the gradient clipping proposed by Pan et al. [2] would effectively resolve both issues. We clarify, however, that such integration falls outside the immediate scope of our current paper.
>
> [2] Pan, Zijie, et al. "Enhancing High-Resolution 3D Generation through Pixel-wise Gradient Clipping." ICLR 2024.
>
> We thank Reviewer tCwW again for the constructive feedback.

---

> > ### Comment · Reviewer_tCwW · 2025-11-27
> >
> > Thank you for responding to my questions. I would like to maintain my score.

---

> > > ### Author Response · Authors · 2025-11-27
> > >
> > > Thank you for your thoughtful review. Your feedback has been very helpful in improving our work and strengthening the manuscript. We will ensure that all your insights are fully reflected in the final version, and please feel free to reach out if there are any additional points to discuss.

---

### Official Review · Reviewer_J8ge · 2025-11-01

**Soundness:** 2
**Presentation:** 2
**Contribution:** 2
**Rating:** 4
**Confidence:** 4

**Summary:**

Although the current mainstream "Score Distillation Sampling" (SDS) method has made significant progress by utilizing 2D diffusion models, it generally suffers from problems such as over-smoothed output results, oversaturated colors, and excessively large update variance. To address these issues, some recent methods have introduced DDIM inversion technology to stabilize the optimization process. However, the authors found that DDIM inversion introduces and accumulates discretization errors, which, after thousands of iterations, lead to serious structural distortions and color distortion in the generated 3D models. The authors proposed a new score distillation framework called "Progressive Latent Calibration" (PLC). PLC abandons DDIM inversion and adopts a multi-step sampling strategy to generate high-quality "pseudo-ground truth" samples to replace the single-step denoising in SDS, thereby obtaining more stable and refined supervision signals.

**Strengths:**

The article is clear and fluent, making it easy to read. The "non-inversion" approach of PLC has opened up a new technical path for solving the stability problem of score distillation. The article clearly reveals its motivation through preliminary experiments in the appendix section. Whether from the perspective of quantitative data or visual effects, this method has achieved excellent results. The qualitative graphs (such as Figure 4) also clearly demonstrate its advantages in color and details.

**Weaknesses:**

- Does the number of sampling steps N have a linearly increasing impact on the memory consumption and computational efficiency of PLC? - - Does the model quality also increase with the increase of N? Why was N=4 chosen as the final number of sampling steps? Are there more examples of Ablation studies?
- There are some detailed questions. Please elaborate on why the auxiliary loss chooses the unconditional prediction after the first update? Why not use conditional prediction? Why wasn't supervised prediction continued afterward? Additionally, why is it matched with the original CFG prediction?

**Questions:**

Please see weakness

---

> ### Author Response · Authors · 2025-11-21
>
> We sincerely thank Reviewer J8ge for the constructive feedback and assessment. We are glad you found our paper clear and fluent, our approach to be a novel and valuable technical path, and our quantitative and qualitative results to be excellent.
>
> We address the weaknesses and questions below.
>
> ---
>
> ### **Weakness 1: Ablation on the Number of Sampling Steps (**$N$**)**
>
> The reviewer asked about the impact of the number of sampling steps, $N$, on computational cost and generated sample quality, and our rationale for choosing $N=4$. We briefly explained our choice in Section 4.4 of the main manuscript, supported by qualitative visualizations in Figure 5. We agree that this quantitative grounding may appear insufficient, and thus, we measured additional quantitative metrics across varying $N$ values as requested. Also, we included this analysis in Appendix L of the revised manuscript.
>
> ### Quantitative Ablation on Sampling Steps ($N$)
>
> | **N (Steps)** | **FID ↓** | **CLIP Score ↑** | **Runtime (min)** |
> | --- | --- | --- | --- |
> | 3 | 108.15 | 32.59 | 104 |
> | 4 (Ours) | **100.41** | 32.68 | 133 |
> | 5 | 102.08 | 32.69 | 150 |
> | 6 | 101.63 | **32.81** | 160 |
>
> As the quantitative results demonstrate, the model exhibits insufficient performance when $N$ is smaller than 4 (i.e., $N=3$), resulting in lower FID and CLIP scores compared to $N \ge 4$.
>
> Conversely, from $N=4$ onward, there are no significant changes in performance in terms of FID or CLIP score. While the runtime increases linearly with $N$, VRAM usage does not increase, as this multi-step process does not allocate additional parameters.
>
> Furthermore, we have included additional visual samples in Figure 12 of the revised manuscript's supplementary material (Appendix L) to address the reviewer's request for further qualitative evidence. As can be seen in these samples, the qualitative performance difference between $N=4, 5, 6$ is minor, aside from a slight increase in detail.
> Therefore, we concluded that $N=4$ provides the optimal trade-off between generation quality and efficiency.
>
> ---
>
> ### **Weakness 2: Rationale for the Auxiliary Loss (**$\mathcal{L}_{aux}$**)**
>
> We structured our auxiliary loss based on the intuition that early regularization of the un-guided sampling path is critical to stabilizing the 3D asset.
>
> Specifically, $\mathcal L_{aux}$ forces the foundational, unconditional prediction ($\bar x_{0,2}^{\phi,\emptyset}$) to align directly with the ideal prompt-guided target ($\hat{x}_{0}^{\phi,y}$). The core intuition is that the unconditional prediction captures the model's general prior belief about the structure. By aligning this core structure with the prompt's semantics right at the initial update step ($n=1$), we establish a robust foundation for the multi-step process. This prevents the base or un-guided structure of the generated pseudo-GT from drifting away from the target semantics, thereby enhancing overall fidelity and stability.
>
> Although the formulation might appear heuristic, this mechanism proves effective, as demonstrated by the resulting enhanced performance in both CLIP Score and FID. Crucially, $\mathcal L_{aux}$ achieves this gain with negligible additional computational overhead, as all necessary intermediate components ($\hat x_0^{\phi,y}$ and $\bar{x}_{0,2}^{\phi,\emptyset}$) are derived from the main loss calculation and require no extra diffusion U-Net inferences.

---

### Author Response · Authors · 2025-11-24
**Revision Summary**

We sincerely thank all reviewers for their constructive feedback and valuable insights. We have uploaded a revised manuscript to reflect your suggestions, with modifications highlighted in blue.

The key updates in the revised manuscript include:

**Ablation on Hyperparameters:** We have added ablation studies to analyze the sensitivity of our method to different hyperparameters. (Appendix L and M, Table 8, Figure 12 and 13)

**Color Fidelity Evaluation:** We have included a quantitative evaluation of color fidelity (saturation and contrast scores) to strictly validate our improvements in preventing color distortion. (Section 4.5, Table 3)

**Updated Related Works:** We have updated the Related Works section to discuss and compare our approach with flow matching-based methods. (Appendix F)

Detailed responses addressing the specific questions and concerns raised by each reviewer have been posted as individual comments in the respective review threads.

Thank you again for your time and effort in reviewing our work.

Best regards,

The Authors

---

### Author Response · Authors · 2025-12-02
**Author Final Remark: Summary of Contributions and Revisions**

To the Area Chair and Reviewers,

We sincerely thank the Area Chair and all reviewers for their time and effort in evaluating our paper. Given the transition to a new Area Chair, we first summarize the reviewers’ main positive assessments of our work, followed by an overview of how we have addressed their concerns through detailed responses and manuscript revisions.

---

**Summary of Positive Feedback**

We are encouraged that the reviewers recognized the value of our work, highlighting the following strengths:

- **Novelty & Motivation:** Reviewers commended the **solid intuitions** behind our method and recognized the non-inversion approach of PLC as a **new technical path** that correctly identifies the root cause of errors in previous inversion-based methods (Reviewers J8ge, tCwW, FiWq, LKaP).
- **Performance:** The quantitative and qualitative results were praised as **excellent**, with specific acknowledgments of our improvements in color fidelity and detail preservation (Reviewers J8ge, FiWq, LKaP).
- **Clarity:** The paper was described as clear, fluent, and well-structured (Reviewers J8ge, LKaP).

---

**Summary of Responses and Revisions**

During the rebuttal period, we have carefully addressed the reviewers' questions and incorporated their feedback into the revised manuscript.

1. **Ablation Studies (Addressing J8ge, tCwW):**
    - We conducted extensive ablations on the number of sampling steps, residual scaling, and denoising schedules.
    - **Revision:** These results are added to **Appendix L and M**, demonstrating that our choice of parameters offers a favorable trade-off between quality and efficiency.
2. **Quantitative Color Fidelity (Addressing FiWq):**
    - To quantitatively validate our claims on color improvement, we measured saturation and contrast scores.
    - **Revision:** We added **Section 4.5**, showing that PLC achieves color statistics closest to real images compared to baselines.
3. **Expanded Related Works (Addressing LKaP):**
    - We clarified the positioning of our work relative to flow matching-based methods (e.g., FlowDreamer, RFDS).
    - **Revision:** A detailed description regarding the scope of flow matching-based methods has been added to **Appendix F**.
4. **Clarifications and Analysis:**
    - We provided detailed rationales for the auxiliary loss formulation (J8ge) and the exclusion of VSD from the user study due to differing 3D representations (FiWq). We also noted that the failure mode analysis was already included in Appendix E of the original manuscript (tCwW).

---

**Closing Remarks regarding the Review Process**

We are fully aware of the recent unprecedented security incident and the subsequent reset of the review process taken by the ICLR program chairs. We understand that due to the reversion of reviews and the prohibition on further reviewer participation, we were **unable to have an interactive discussion or receive final feedback** from Reviewers J8ge, FiWq, and LKaP regarding our responses. While it is unfortunate that we could not engage further with these reviewers to confirm that their concerns were fully resolved, **we believe our detailed responses and the substantive updates in the revised manuscript effectively address the concerns they raised.** We respectfully hope that these clarifications and substantive updates, together with the strong consensus on our method’s novelty and effectiveness, will be taken into full consideration for the final decision.

---

Sincerely,

The Authors

---

### Meta-Review · Area_Chair_v5Qg · 2026-01-07

**Summary:**

The paper proposes PLC, an inversion-free score distillation framework for text-to-3D. It argues repeated DDIM inversion causes discretization error accumulation that leads to structural and color artifacts, and replaces inversion with multi-step pseudo-ground-truth sampling plus a latent calibration residual to reduce gradient variance. Strengths: clear motivation, strong qualitative results, good quantitative gains over SDS and inversion-based baselines, and added color-fidelity measurements and ablations in the revision. However, the core mechanism is still largely heuristic and somewhat incremental, compute cost scales with sampling steps, multi-view consistency issues remain, and the claimed efficiency advantage is not fully convincing.

**Reviewer Concerns:**

Addressed in rebuttal/revision: ablations on sampling steps and key hyperparameters, added quantitative color metrics (saturation/contrast), clarified auxiliary loss rationale, explained why VSD was excluded from the user study, and expanded related-work positioning (including flow-matching).

Still outstanding: limited scope/novelty, cost and cost framing, lack of multi-view consistency treatment, and unclear generalization to harder diffusion backbones (for example SDXL).

**Reviewer Scores:**

Confirmed scores: J8ge = 4 (below threshold), tCwW = 6, FiWq = 6, LKaP = 6 (three marginally above threshold). tCwW explicitly stated they would keep their score after the rebuttal. The other reviewers did not provide post-rebuttal score updates; at best, J8ge might increase slightly with the added ablations, while FiWq and LKaP likely stay similar given remaining novelty and cost concerns.

---

### Decision · Program_Chairs · 2026-01-26

Reject